# LEARN TO EXPLAIN EFFICIENTLY VIA NEURAL LOGIC INDUCTIVE LEARNING

**Yuan Yang & Le Song**
Georgia Institute of Technology
`yyang754@gatech.edu, lsong@cc.gatech.edu`

## ABSTRACT

The capability of making interpretable and self-explanatory decisions is essential for developing responsible machine learning systems. In this work, we study the learning to explain problem in the scope of inductive logic programming (ILP). We propose Neural Logic Inductive Learning (NLIL), an efficient differentiable ILP framework that learns first-order logic rules that can explain the patterns in the data. In experiments, compared with the state-of-the-art methods, we find NLIL can search for rules that are x10 times longer while remaining x3 times faster. We also show that NLIL can scale to large image datasets, i.e. Visual Genome, with 1M entities.

## 1 INTRODUCTION

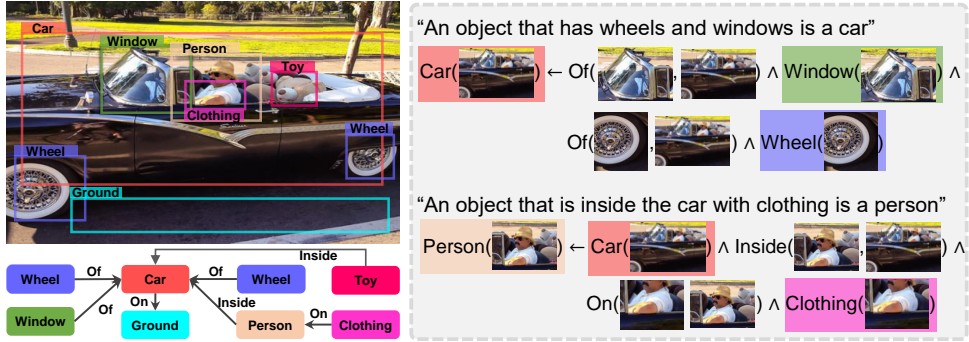

Figure 1: A scene-graph can describe the relations of objects in an image. The NLIL can utilize this graph and explain the presence of objects `Car` and `Person` by learning the first-order logic rules that characterize the common sub-patterns in the graph. The explanation is globally consistent and can be interpreted as commonsense knowledge.

The recent years have witnessed the growing success of deep learning models in a wide range of applications. However, these models are also criticized for the lack of interpretability in its behavior and decision making process (Lipton, 2016; Mittelstadt et al., 2019), and for being data-hungry. The ability to explain its decision is essential for developing a responsible and robust decision system (Guidotti et al., 2019). On the other hand, logic programming methods, in the form of first-order logic (FOL), are capable of discovering and representing knowledge in explicit symbolic structure that can be understood and examined by human (Evans & Grefenstette, 2018).

In this paper, we investigate the learning to explain problem in the scope of inductive logic programming (ILP) which seeks to learn first-order logic rules that explain the data. Traditional ILP methods (Galárraga et al., 2015) rely on hard matching and discrete logic for rule search which is not tolerant for ambiguous and noisy data (Evans & Grefenstette, 2018). A number of works are proposed for developing differentiable ILP models that combine the strength of neural and logic-based computation (Evans & Grefenstette, 2018; Campero et al., 2018; Rocktäschel & Riedel, 2017; Payani & Fekri, 2019; Dong et al., 2019). Methods such as ∂ILP (Evans & Grefenstette, 2018) are referred to as forward-chaining methods. It constructs rules using a set of pre-defined templates and

evaluates them by applying the rule on *background data* multiple times to deduce new facts that lie in the held-out set (related works available at Appendix A). However, general ILP problem involves several steps that are NP-hard: (i) the rule search space grows exponentially in the length of the rule; (ii) assigning the logic variables to be shared by predicates grows exponentially in the number of arguments, which we refer as variable binding problem; (iii) the number of rule *instantiations* needed for formula evaluation grows exponentially in the size of data. To alleviate these complexities, most works have limited the search length to within 3 and resort to template-based variable assignments, limiting the expressiveness of the learned rules (detailed discussion available at Appendix B). Still, most of the works are limited in small scale problems with less than 10 relations and 1K entities.

On the other hand, multi-hop reasoning methods (Guu et al., 2015; Lao & Cohen, 2010; Lin et al., 2015; Gardner & Mitchell, 2015; Das et al., 2016) are proposed for the knowledge base (KB) completion task. Methods such as NeuralLP (Yang et al., 2017) can answer the KB queries by searching for a relational path that leads from the subject to the object. These methods can be interpreted in the ILP domain where the learned relational path is equivalent to a chain-like first-order rule. Compared to the template-based counterparts, methods such as NeuralLP is highly efficient in variable binding and rule evaluation. However, they are limited in two aspects: (i) the chain-like rules represent a subset of the Horn clauses, and are limited in expressing complex rules such as those shown in Figure 1; (ii) the relational path is generated while conditioning on the specific query, meaning that the learned rule is only valid for the current query. This makes it difficult to learn rules that are globally consistent in the KB, which is an important aspect of a good explanation.

In this work, we propose Neural Logic Inductive Learning (NLIL), a differentiable ILP method that extends the multi-hop reasoning framework for general ILP problem. NLIL is highly efficient and expressive. We propose a divide-and-conquer strategy and decompose the search space into 3 subspaces in a hierarchy, where each of them can be searched efficiently using attentions. This enables us to search for x10 times longer rules while remaining x3 times faster than the state-of-the-art methods. We maintain the global consistency of rules by splitting the training into rule generation and rule evaluation phase, where the former is only conditioned on the predicate type that is shared globally.

And more importantly, we show that a scalable ILP method is widely applicable for model explanations in supervised learning scenario. We apply NLIL on Visual Genome (Krishna et al., 2016) dataset for learning explanations for 150 object classes over 1M entities. We demonstrate that the learned rules, while maintaining the interpretability, have comparable predictive power as densely supervised models, and generalize well with less than 1% of the data.

## 2 PRELIMINARIES

Supervised learning typically involves learning classifiers that map an object from its input space to a score between 0 and 1. How can one explain the outcome of a classifier? Recent works on interpretability focus on generating heatmaps or attention that self-explains a classifier (Ribeiro et al., 2016; Chen et al., 2018; Olah et al., 2018). We argue that a more effective and human-intelligent explanation is through the description of the connection with other classifiers.

For example, consider an object detector with classifiers $\texttt{Person}(X)$, $\texttt{Car}(X)$, $\texttt{Clothing}(X)$ and $\texttt{Inside}(X, X')$ that detects if certain region contains a person, a car, a clothing or is inside another region, respectively. To explain why a person is present, one can leverage its connection with other attributes, such as "$X$ is a person if it's inside a car and wearing clothing", as shown in Figure 1. This intuition draws a close connection to a longstanding problem of first-order logic literature, i.e. **Inductive Logic Programming** (ILP).

### 2.1 INDUCTIVE LOGIC PROGRAMMING

A typical first-order logic system consists of 3 components: **entity**, **predicate** and **formula**. Entities are objects $\mathbf{x} \in \mathcal{X}$. For example, for a given image, a certain region is an entity $\mathbf{x}$, and the set of all possible regions is $\mathcal{X}$. Predicates are functions that map entities to 0 or 1, for example $\texttt{Person} : \mathbf{x} \mapsto \{0, 1\}$, $\mathbf{x} \in \mathcal{X}$. Classifiers can be seen as *soft* predicates. Predicates can take multiple arguments, e.g. $\texttt{Inside}$ is a predicate with 2 inputs. The number of arguments is referred to as the *arity*. **Atom** is a predicate symbol applied to a logic variable, e.g. $\texttt{Person}(X)$ and $\texttt{Inside}(X, X')$. A logic variable such as $X$ can be **instantiated** into any object in $\mathcal{X}$.

A first-order logic (FOL) formula is a combination of atoms using logical operations $\{\wedge, \vee, \neg\}$ which correspond to logic `and`, `or` and `not` respectively. Given a set of predicates $\mathcal{P} = \{P_1, ..., P_K\}$, we define the explanation of a predicate $P_k$ as a first-order logic entailment

$$\forall X, X' \, \exists Y_1, Y_2... \, P_k(X, X') \leftarrow A(X, X', Y_1, Y_2...), \qquad (1)$$

where $P_k(X, X')$ is the *head* of the entailment, and it will become $P_k(X)$ if it is a unary predicate. $A$ is defined as the rule *body* and is a general formula, e.g. conjunction normal form (CNF), that is made of atoms with predicate symbols from $\mathcal{P}$ and logic variables that are either head variables $X$, $X'$ or one of the body variables $\mathcal{Y} = \{Y_1, Y_2, ...\}$.

By using the logic variables, the explanation becomes transferrable as it represents the "lifted" knowledge that does not depend on the specific data. It can be easily interpreted. For example,

$$\texttt{Person}(X) \leftarrow \texttt{Inside}(X, Y_1) \wedge \texttt{Car}(Y_1) \wedge \texttt{On}(Y_2, X) \wedge \texttt{Clothing}(Y_2) \qquad (2)$$

represents the knowledge that "if an object is inside the car with clothing on it, then it's a person". To evaluate a formula on the actual data, one **grounds** the formula by instantiating all the variables into objects. For example, in Figure 1, Eq.(2) is applied to the specific regions of an image.

Given a relational knowledge base (KB) that consists of a set of facts $\{\langle \mathbf{x}_i, P_i, \mathbf{x}'_i \rangle\}_{i=1}^N$ where $P_i \in \mathcal{P}$ and $\mathbf{x}_i, \mathbf{x}'_i \in \mathcal{X}$. The task of learning FOL rules in the form of Eq.(1) that entail target predicate $P^* \in \mathcal{P}$ is called inductive logic programming. For simplicity, we consider unary and binary predicates for the following contents, but this definition can be extended to predicates with higher arity as well.

## 2.2 MUTLI-HOP REASONING

The ILP problem is closely related to the multi-hop reasoning task on the knowledge graph (Guu et al., 2015; Lao & Cohen, 2010; Lin et al., 2015; Gardner & Mitchell, 2015; Das et al., 2016). Similar to ILP, the task operates on a KB that consists of a set of predicates $\mathcal{P}$. Here the facts are stored with respect to the predicate $P_k$ which is represented as a binary matrix $\mathbf{M}_k$ in $\{0, 1\}^{|\mathcal{X}| \times |\mathcal{X}|}$. This is an adjacency matrix, meaning that $\langle \mathbf{x}_i, P_k, \mathbf{x}_j \rangle$ is in the KB if and only if the $(i, j)$ entry of $\mathbf{M}_k$ is 1.

Given a query $q = \langle \mathbf{x}, P^*, \mathbf{x}' \rangle$. The task is to find a relational path $\mathbf{x} \xrightarrow{P^{(1)}} ... \xrightarrow{P^{(T)}} \mathbf{x}'$, such that the two query entities are connected. Formally, let $\mathbf{v}_\mathbf{x}$ be the one-hot encoding of object $\mathbf{x}$ with dimension of $|\mathcal{X}|$. Then, the $(t)$th hop of the reasoning along the path is represented as

$$\mathbf{v}^{(0)} = \mathbf{v}_\mathbf{x}, \qquad\qquad \mathbf{v}^{(t)} = \mathbf{M}^{(t)} \mathbf{v}^{(t-1)},$$

where $\mathbf{M}^{(t)}$ is the adjacency matrix of the predicate used in $(t)$th hop. The $\mathbf{v}^{(t)}$ is the *path features vector*, where the $j$th element $v_j^{(t)}$ counts the number of unique paths from $\mathbf{x}$ to $\mathbf{x}_j$ (Guu et al., 2015). After $T$ steps of reasoning, the score of the query is computed as

$$\text{score}(\mathbf{x}, \mathbf{x}') = \mathbf{v}_{\mathbf{x}'}^\top \prod_{t=1}^T \mathbf{M}^{(t)} \cdot \mathbf{v}_\mathbf{x}. \qquad (3)$$

For each $q$, the goal is to (i) find an appropriate $T$ and (ii) for each $t \in [1, 2, ..., T]$, find the appropriate $\mathbf{M}^{(t)}$ to multiply, such that Eq.(3) is maximized. These two discrete picks can be relaxed as learning the weighted sum of scores from all possible paths, and weighted sum of matrices at each step. Let

$$\kappa(\mathbf{s}_\psi, \mathbf{S}_\varphi) \equiv \sum_{t'=1}^T s_\psi^{(t')} \left( \prod_{t=1}^{t'} \sum_{k=1}^K s_{\varphi,k}^{(t)} \mathbf{M}_k \right) \qquad (4)$$

be the *soft path selection function* parameterized by (i) the *path attention vector* $\mathbf{s}_\psi = [s_\psi^{(1)}, ..., s_\psi^{(T)}]^\top$ that softly picks the best path with length between 1 to T that answers the query, and (ii) the *operator attention vectors* $\mathbf{S}_\varphi = [\mathbf{s}_\varphi^{(1)}, ..., \mathbf{s}_\varphi^{(T)}]^\top$, where $\mathbf{s}_\varphi^{(t)}$ softly picks the $\mathbf{M}^{(t)}$ at $(t)$th step. Here we omit the dependence on $M_k$ for notation clarity. These two attentions are generated with a model

$$\mathbf{s}_\psi, \mathbf{S}_\varphi = \mathbb{T}(\mathbf{x}; \mathbf{w}) \qquad (5)$$

with learnable parameters $\mathbf{w}$. For methods such as (Guu et al., 2015; Lao & Cohen, 2010), $\mathbb{T}(\mathbf{x}; \mathbf{w})$ is a random walk sampler which generates one-hot vectors that simulate the random walk on the graph starting from $\mathbf{x}$. And in NeuralLP (Yang et al., 2017), $\mathbb{T}(\mathbf{x}; \mathbf{w})$ is an RNN controller that generates a sequence of normalized attention vectors with $\mathbf{v}_\mathbf{x}$ as the initial input. Therefore, the objective is defined as

$$\arg\max_{\mathbf{w}} \sum_q \mathbf{v}_{\mathbf{x}'}^\top \kappa(\mathbf{s}_\psi, \mathbf{S}_\varphi) \mathbf{v}_\mathbf{x}, \tag{6}$$

Learning the relational path in the multi-hop reasoning can be interpreted as solving an ILP problem with chain-like FOL rules (Yang et al., 2017)

$$P^*(X, X') \leftarrow P^{(1)}(X, Y_1) \land P^{(2)}(Y_1, Y_2) \land \dots \land P^{(T)}(Y_{n-1}, X').$$

Compared to the template-based ILP methods such as $\partial$ILP, this class of methods is efficient in rule exploration and evaluation. However, **(P1)** generating explanations for supervised models puts a high demand on the rule expressiveness. The chain-like rule space is limited in its expressive power because it represents a constrained subspace of the Horn clauses rule space. For example, Eq.(2) is a Horn clause and is not chain-like. And the ability to efficiently search beyond the chain-like rule space is still lacking in these methods. On the other hand, **(P2)** the attention generator $\mathbb{T}(\mathbf{x}; \mathbf{w})$ is dependent on $\mathbf{x}$, the subject of a specific query $q$, meaning that the explanation generated for target $P^*$ can vary from query to query. This makes it difficult to learn FOL rules that are globally consistent in the KB.

## 3 NEURAL LOGIC INDUCTIVE LEARNING

In this section, we show the connection between the multi-hop reasoning methods with the general logic entailment defined in Eq.(1). Then we propose a hierarchical rule space to solve **(P1)**, i.e. we extend the chain-like space for efficient learning of more expressive rules.

### 3.1 THE OPERATOR VIEW

In Eq.(1), variables that only appear in the body are under existential quantifier. We can turn Eq.(1) into *Skolem normal form* by replacing all variables under existential quantifier with functions with respect to $X$ and $X'$,

$$\forall X, X' \exists \varphi_1, \varphi_2, \dots \; P^*(X, X') \leftarrow A(X, X', \varphi_1(X), \varphi_1(X'), \varphi_2(X), \dots). \tag{7}$$

If the functions are known, Eq.(7) will be much easier to evaluate than Eq.(1). Because grounding this formula only requires to instantiate the head variables, and the rest of the body variables are then determined by the deterministic functions.

Functions in Eq.(7) can be arbitrary. But what are the functions that one can utilize? We propose to adopt the notions in section 2.2 and treat each predicate as an *operator*, such that we have a subspace of the functions $\Phi = \{\varphi_1, \dots, \varphi_K\}$, where

$$\begin{cases} \varphi_k() = \mathbf{M}_k \, \mathbf{1} & \text{if } k \in \mathcal{U}, \\ \varphi_k(\mathbf{v}_\mathbf{x}) = \mathbf{M}_k \mathbf{v}_\mathbf{x} & \text{if } k \in \mathcal{B}, \end{cases}$$

where $\mathcal{U}$ and $\mathcal{B}$ are the sets of unary and binary predicates respectively. The operator of the unary predicate takes no input and is parameterized with a diagonal matrix. Intuitively, given a subject entity $\mathbf{x}$, $\varphi_k$ returns the *set embedding* (Guu et al., 2015) that represents the object entities that, together with the subject, satisfy the predicate $P_k$. For example, let $\mathbf{v}_\mathbf{x}$ be the one-hot encoding of an object in the image, then $\varphi_{\texttt{Inside}}(\mathbf{v}_\mathbf{x})$ returns the objects that spatially contain the input box. For unary predicate such as $\texttt{Car}(X)$, its operator $\varphi_{\texttt{Car}}() = \mathbf{M}_{\texttt{car}} \mathbf{1}$ takes no input and returns the set of all objects labelled as car.

Since we only use $\Phi$, a subspace of the functions, the existential variables that can be represented by the operator calls, denoted as $\hat{\mathcal{Y}}$, also form the subset $\hat{\mathcal{Y}} \subseteq \mathcal{Y}$. This is slightly constrained from Eq.(1). For example, in $\texttt{Person}(X) \leftarrow \texttt{Car}(Y)$, $Y$ can not be interpreted as the operator call from $X$. However, we argue that such rules are generally trivial. For example, it's not likely to infer "an image contains a person" by simply checking if "there is any car in the image".

Therefore, any FOL formula that complies with Eq.(7) can now be converted into the operator form and vice versa. For example, Eq.(2) can be written as

$$\texttt{Person}(X) \leftarrow \texttt{Car}(\varphi_{\texttt{Inside}}(X)) \wedge \texttt{On}(\varphi_{\texttt{Clothing}}(), X), \tag{8}$$

where the variable $Y_1$ and $Y_2$ are eliminated. Note that this conversion is not unique. For example, $\texttt{Car}(\varphi_{\texttt{Inside}}(X))$ can be also written as $\texttt{Inside}(X, \varphi_{\texttt{Car}}())$. The variable binding problem now becomes equivalent to the path-finding problem in section 2.2, where one searches for the appropriate chain of operator calls that can represent the variable in $\hat{\mathcal{Y}}$.

## 3.2 PRIMITIVE STATEMENTS

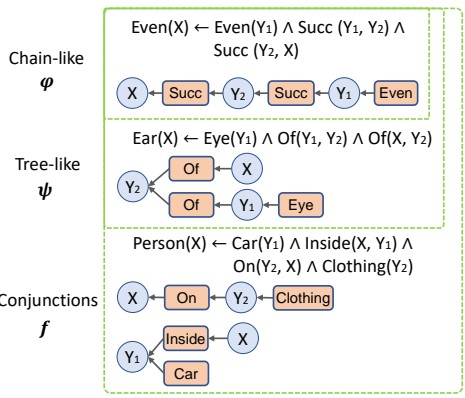

Figure 2: Factor graphs of example chain-like, tree-like and conjunctions of rules. Each rule type is the subset of the latter. Succ stands for successor.

As discussed above, the Eq.(3) is equivalent to a chain-like rule. We want to extend this notion and be able to represent more expressive rules. To do this, we introduce the notion of **primitive statement** $\psi$. Note that an *atom* is defined as a predicate symbol applied to specific logic variables. Similarly, we define a predicate symbol applied to the head variables or those in $\hat{\mathcal{Y}}$ as a primitive statement. For example, in Eq.(8), $\psi_1 = \texttt{Car}(\varphi_{\texttt{Inside}}(X))$ and $\psi_2 = \texttt{On}(\varphi_{\texttt{Clothing}}(), X)$ are two primitive statements.

Similar to an atom, each primitive statement is a mapping from the input space to a scalar confidence score, i.e. $\psi : \mathbb{R}^{|\mathcal{X}|} \times \mathbb{R}^{|\mathcal{X}|} \mapsto s \in [0, 1]$. Formally, for a unary primitive statement $P_k(\varphi^{(T')} \cdot ... \cdot \varphi^{(1)}(\mathbf{x}'))$ and a binary one $P_k(\varphi^{(T)} \cdot ... \cdot \varphi^{(1)}(\mathbf{x}), \varphi^{(T')} \cdot ... \cdot \varphi^{(1)}(\mathbf{x}'))$, their mappings are defined as

$$\psi_k(\mathbf{x}, \mathbf{x}') = \begin{cases} \sigma((\mathbf{M}_k \mathbf{1})^\top (\prod_{t'=1}^{T'} \mathbf{M}^{(t')} \mathbf{v}_{\mathbf{x}'})) & \text{if } k \in \mathcal{U}, \\ \sigma((\mathbf{M}_k \prod_{t=1}^{T} \mathbf{M}^{(t)} \mathbf{v}_{\mathbf{x}})^\top (\prod_{t'=1}^{T'} \mathbf{M}^{(t')} \mathbf{v}_{\mathbf{x}'})) & \text{if } k \in \mathcal{B}, \end{cases} \tag{9}$$

where $\sigma(\cdot)$ is the sigmoid function. Note that we give unary $\psi$ a dummy input $\mathbf{x}$ for notation convenience. For example, in

$$\texttt{Ear}(X) \leftarrow \texttt{Eye}(Y_1) \wedge \texttt{Of}(Y_1, Y_2) \wedge \texttt{Of}(X, Y_2),$$

the body is a single statement $\psi = \texttt{Of}(\varphi_{\texttt{Eye}}(), \varphi_{\texttt{Of}}(X))$. Its value is computed as $\psi_{\texttt{Of}}(\varphi_{\texttt{Eye}}(), \varphi_{\texttt{Of}}(\mathbf{v}_{\mathbf{x}'})) = \sigma((\mathbf{M}_{\texttt{Of}} \mathbf{M}_{\texttt{Eye}} \mathbf{1})^\top (\mathbf{M}_{\texttt{Of}} \mathbf{v}_{\mathbf{x}'}))$. Compared to Eq.(3), Eq.(9) replaces the target $\mathbf{v}_{\mathbf{x}'}$ into another relational path. This makes it possible to represent "correlations" between two variables, and the path that starts from the unary operator, e.g. $\varphi_{\texttt{Eye}}()$. To see this, one can view a FOL rule as a factor graph with logic variables as the nodes and predicates as the potentials (Cohen et al., 2017). And running the operator call is essentially conducting the belief propagation over the graph in a fixed direction. As shown in Figure 2, primitive statement is capable of representing the tree-like factor graphs, which significantly improves the expressive power of the learned rules.

Similarly, Eq.(9) can be relaxed into weighted sums. In Eq.(6), all relational paths are summed with a single path attention vector $\mathbf{s}_\psi$. We extend this notion by assigning separate vectors for each argument of the statement $\psi$. Let $\mathbf{S}_\psi, \mathbf{S}'_\psi \in \mathbb{R}^{K \times T}$ be the path attention matrices for the first and second argument of all statements in $\Psi$, i.e. $\mathbf{s}_{\psi,k}$ and $\mathbf{s}'_{\psi,k}$ are the path attention vectors of the first and second argument of the $k$th statement. Then we have

$$\psi_k(\mathbf{x}, \mathbf{x}') = \begin{cases} \sigma\left((\mathbf{M}_k \mathbf{1})^\top (\kappa(\mathbf{s}'_{\psi,k}, \mathbf{S}_\varphi) \mathbf{v}_{\mathbf{x}'})\right) & \text{if } k \in \mathcal{U}, \\ \sigma\left((\mathbf{M}_k \kappa(\mathbf{s}_{\psi,k}, \mathbf{S}_\varphi) \mathbf{v}_{\mathbf{x}})^\top (\kappa(\mathbf{s}'_{\psi,k}, \mathbf{S}_\varphi) \mathbf{v}_{\mathbf{x}'})\right) & \text{if } k \in \mathcal{B}. \end{cases} \tag{10}$$

## 3.3 LOGIC COMBINATION SPACE

By introducing the primitive statements, we are now one step away from representing the running example rule Eq.(8), which is the logic conjunction of two statements $\psi_1$ and $\psi_2$. Specifically,

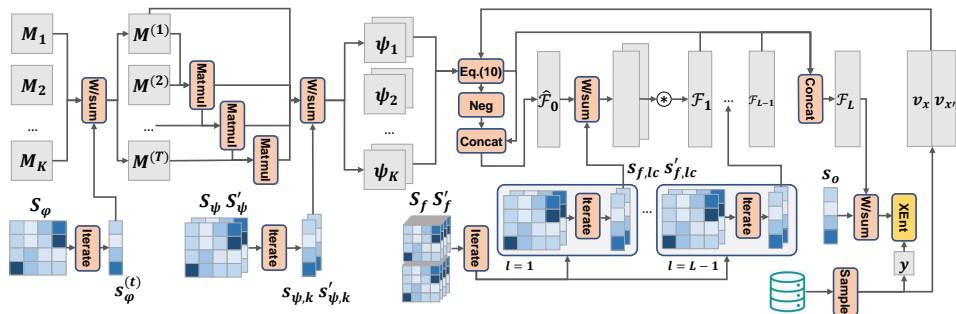

Figure 3: A hierarchical rule space where the operator calls, statement evaluations and logic combinations are all relaxed into the weight sums with respect to attentions $\mathbf{S}_\varphi, \mathbf{S}_\psi, \mathbf{S}'_\psi, \mathbf{S}_f, \mathbf{S}'_f$ and $\mathbf{s}_o$. W/sum denotes the weighted sum, Matmul denotes the matrix product, Neg denotes soft logic not, and XEnt denotes the cross-entropy loss.

we want to further extend the rule search space by exploring the logic combinations of primitive statements, via $\{\wedge, \vee, \neg\}$, as shown in Figure 2. To do this, we utilize the soft logic not and soft logic and operations

$$\neg p = 1 - p, \qquad\qquad p \wedge q = p * q,$$

where $p, q \in [0, 1]$. Here we do not include the logic $\vee$ operation because it can be implicitly represented as $p \vee q = \neg(\neg p \wedge \neg q)$. Let $\Psi = \{\psi_k(\mathbf{x}, \mathbf{x}')\}_{k=1}^K$ be the set of primitive statements with all possible predicate symbols. We define the *formula set* at $l$th level as

$$\begin{aligned} \mathcal{F}_0 &= \Psi, \\ \hat{\mathcal{F}}_{l-1} &= \mathcal{F}_{l-1} \cup \{1 - f(\mathbf{x}, \mathbf{x}') : f \in \mathcal{F}_{l-1}\}, \\ \mathcal{F}_l &= \{f_i(\mathbf{x}, \mathbf{x}') * f'_i(\mathbf{x}, \mathbf{x}') : f_i, f'_i \in \hat{\mathcal{F}}_{l-1}\}_{i=1}^C, \end{aligned} \qquad (11)$$

where each element in the formula set $\{f : f \in \mathcal{F}_l\}$ is called a formula such that $f : \mathbb{R}^{|\mathcal{X}|} \times \mathbb{R}^{|\mathcal{X}|} \mapsto s \in [0, 1]$. Intuitively, we define the logic combination space in a similar way as that in pathfinding: the initial formula set contains only primitive statements $\Psi$, because they are formulas by themselves. For the $l - 1$th formula set $\mathcal{F}_{l-1}$, we concatenate it with its logic negation, which yields $\hat{\mathcal{F}}_{l-1}$. Then each formula in the next level is the logic and of two formulas from $\hat{\mathcal{F}}_{l-1}$. Enumerating all possible combinations at each level is expensive, so we set up a memory limitation $C$ to indicate the maximum number of combinations each level can keep track of[1]. In other words, each level $\mathcal{F}_l$ is to search for $C$ logic and combinations on formulas from the previous level $\hat{\mathcal{F}}_{l-1}$, such that the $c$th formula at the $l$th level $f_{lc}$ is

$$f_{lc}(\mathbf{x}, \mathbf{x}') = f_{l-1,i}(\mathbf{x}, \mathbf{x}') * f'_{l-1,i}(\mathbf{x}, \mathbf{x}'), \qquad f_{l-1,i}, f'_{l-1,i} \in \hat{\mathcal{F}}_{l-1}. \qquad (12)$$

As an example, for $\Psi = \{\psi_1, \psi_2\}$ and $C = 2$, one possible level sequence is $\mathcal{F}_0 = \{\psi_1, \psi_2\}$, $\mathcal{F}_1 = \{\psi_1 * \psi_2, (1 - \psi_2) * \psi_1\}$, $\mathcal{F}_2 = \{(\psi_1 * \psi_2) * ((1 - \psi_2) * \psi_1), ...\}$ and etc. To collect the rules from all levels, the final level $L$ is the union of previous sets, i.e. $\mathcal{F}_L = \mathcal{F}_0 \cup ... \cup \mathcal{F}_{L-1}$. Note that Eq.(11) does not explicitly forbid trivial rules such as $\psi_1 * (1 - \psi_1)$ that is always true regardless of the input. This is alleviated by introducing nonexistent queries during the training (detailed discussion at section 5).

Again, the rule selection can be parameterized into the weighted-sum form with respect to the attentions. We define the *formula attention tensors* as $\mathbf{S}_f, \mathbf{S}'_f \in \mathbb{R}^{L-1 \times C \times 2C}$, such that $f_{lc}$ is the product of two summations over the previous outputs weighted by attention vectors $\mathbf{s}_{f,lc}$ and $\mathbf{s}'_{f,lc}$ respectively[2]. Formally, we have

$$f_{lc}(\mathbf{x}, \mathbf{x}') = \mathbf{s}_{f,lc}^\top \mathbf{f}_{l-1}(\mathbf{x}, \mathbf{x}') * \mathbf{s}'^\top_{f,lc} \mathbf{f}_{l-1}(\mathbf{x}, \mathbf{x}'), \qquad (13)$$

where $\mathbf{f}_{l-1}(\mathbf{x}, \mathbf{x}') \in \mathbb{R}^{2C}$ is the stacked outputs of all formulas $f \in \hat{\mathcal{F}}_{l-1}$ with arguments $\langle \mathbf{x}, \mathbf{x}' \rangle$. Finally, we want to select the best explanation and compute the score for each query. Let $\mathbf{s}_o$ be the

---

[1] $C$ can vary from level to level, and we keep it the same for notation simplicity

[2] Formula set at $(0)$th level $\hat{\mathcal{F}}_0$ actually contains $2K$ formulas. Here we assume $C = K$ for notation simplicity.

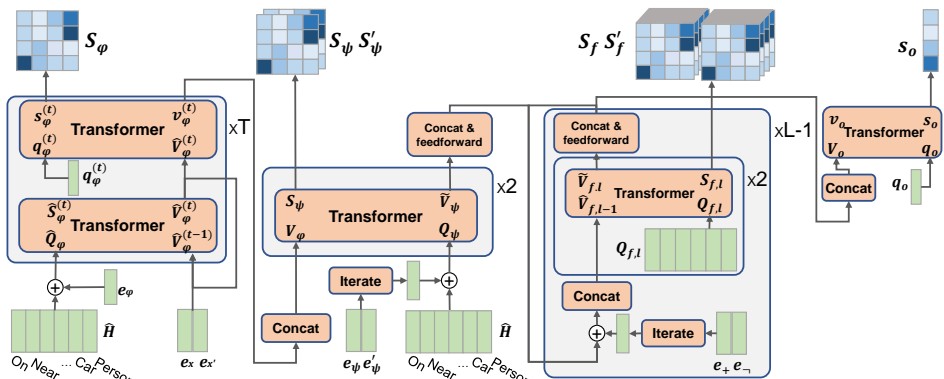

Figure 4: The hierarchical Transformer networks for attention generation without conditioning on the query.

attention vector over $\mathcal{F}_L$, so the output score is defined as

$$\text{score}(\mathbf{x}, \mathbf{x}') = \mathbf{s}_o^\top \mathbf{f}_L(\mathbf{x}, \mathbf{x}').$$  (14)

An overview of the relaxed hierarchical rule space is illustrated in Figure 3.

## 4  HIERARCHICAL TRANSFORMER NETWORKS FOR RULE GENERATION

We have defined a hierarchical rule space as shown in Figure 3, where the discrete selections on the operators, statements and logic combinations are all relaxed into the weight sums with respect to a series of attention parameters $\mathbf{S}_\varphi, \mathbf{S}_\psi, \mathbf{S}'_\psi, \mathbf{S}_f, \mathbf{S}'_f$ and $\mathbf{s}_o$. In this section, we solve (**P2**), i.e. we propose a differentiable model that generates these attentions without conditioning on the specific query.

The goal of NLIL is to generate data-independent FOL rules. In other words, for each target predicate $P^*$, its rule set $\mathcal{F}_L$ and the final output rule should remain unchanged for all the queries $q = \langle \mathbf{x}, P^*, \mathbf{x}' \rangle$ (which is different from that in Eq.(5)). To do this, we define the learnable embeddings of all predicates as $\mathbf{H} = [\mathbf{h}_1, .., \mathbf{h}_K]^\top \in \mathbb{R}^{K \times d}$, and the embeddings for the "dummy" arguments $X$ and $X'$ as $\mathbf{e}_X, \mathbf{e}_{X'} \in \mathbb{R}^d$. We define the attention generation model as

$$\mathbf{S}_\varphi, \mathbf{S}_\psi, \mathbf{S}'_\psi, \mathbf{S}_f, \mathbf{S}'_f, \mathbf{s}_o = \mathbb{T}(\mathbf{e}_X, \mathbf{h}^*, \mathbf{e}_{X'}; \mathbf{w}),$$  (15)

where $\mathbf{h}^*$ is the embedding of $P^*$, such that attentions only vary with respect to $P^*$.

As shown in Figure 4, we propose a stack of three Transformer (Vaswani et al., 2017) networks for attention generator $\mathbb{T}$. Each module is designed to mimic the actual evaluation that could happen during the operator call, primitive statement evaluation and formula computation respectively with neural networks and "dummy" embeddings. And the attention matrices generated during this simulated evaluation process are kept for evaluating Eq.(14). A MultiHeadAttn is a standard Transformer module such that MultiHeadAttn : $\mathbf{Q}^{q \times d} \times \mathbf{V}^{v \times d} \mapsto \mathbf{O}^{q \times d} \times \mathbf{S}^{q \times v}$, where $d$ is the latent dimension and $q, v$ are the query and value dimensions respectively. It takes the *query* $\mathbf{Q}$ and input value $\mathbf{V}$ (which will be internally transformed into *keys* and *values*), and returns the output value $\mathbf{O}$ and attention matrix $\mathbf{S}$. Intuitively, $\mathbf{S}$ encodes the "compatibility" between query and the value, and $\mathbf{O}$ represents the "outcome" of a query given its compatibility with the input.

**Operator search**: For target predicate $P^*$, we alter the embedding matrix $\mathbf{H}$ with

$$\hat{\mathbf{h}}_k = \text{FeedForward}(\text{Concat}(\mathbf{h}_k, \mathbf{h}^*)), \qquad\qquad \hat{\mathbf{H}} = [\hat{\mathbf{h}}_1, ..., \hat{\mathbf{h}}_K]^\top,$$

such that the rule generation is predicate-specific. Let $\mathbf{q}_\varphi^{(t)}$ be the learnable $t$th step operator query embedding. The operator transformer module is parameterized as

$$\hat{\mathbf{V}}_\varphi^{(0)} = [\mathbf{e}_X, \mathbf{e}_{X'}]^\top, \qquad\qquad \hat{\mathbf{Q}}_\varphi = \hat{\mathbf{H}} + \mathbf{e}_\varphi,$$

$$\hat{\mathbf{V}}_\varphi^{(t)}, \hat{\mathbf{S}}_\varphi^{(t)} = \text{MultiHeadAttn}(\hat{\mathbf{Q}}_\varphi, \hat{\mathbf{V}}_\varphi^{(t-1)}), \qquad \mathbf{v}_\varphi^{(t)}, \mathbf{s}_\varphi^{(t)} = \text{MultiHeadAttn}(\mathbf{q}_\varphi^{(t)}, \hat{\mathbf{V}}_\varphi^{(t)}).$$

Here, $\hat{\mathbf{V}}_\varphi^{(0)}$ is the dummy input embedding representing the starting points of the paths. $\mathbf{e}_\varphi$ is a learnable operator encoding such that $\hat{\mathbf{Q}}_\varphi$ represents the embeddings of all operators $\Phi$. Therefore,

Table 1: MRR, Hits@10 and time (mins) of KB completion tasks.

| Model | FB15K-237 | | | WN18 | | |
| --- | --- | --- | --- | --- | --- | --- |
| | MRR | Hits@10 | Time | MRR | Hits@10 | Time |
| NeuralLP | 0.24 | 36.2 | 250 | 0.94 | 94.5 | 54 |
| TransE | 0.28 | 44.5 | **35** | 0.57 | 93.3 | 53 |
| RotatE | **0.34** | **52.6** | 342 | 0.94 | **95.5** | 254 |
| NLIL | 0.25 | 32.4 | 82 | **0.95** | 94.6 | **12** |

Table 2: Statistics of benchmark KBs and Visual Genome scene-graphs.

| KB | # facts | # entities | # predicates |
| --- | --- | --- | --- |
| ES-10 | 17 | 10 | 3 |
| ES-50 | 77 | 50 | 3 |
| ES-1K | 1.5K | 1K | 3 |
| WN18 | 106K | 40K | 18 |
| FB15K | 272K | 15K | 237 |
| VG | 1.9M | 1.4M | 2100 |

we consider that $\hat{\mathbf{V}}_\varphi^{(t)}$ encodes the outputs of the operator calls of $K$ predicates. And we aggregate the outputs with another $\mathrm{MultiHeadAttn}$ with respect to a single query $\mathbf{q}_\varphi^{(t)}$, which in turn yields the operator path attention vector $\mathbf{s}_\varphi^{(t)}$ and aggregated output $\mathbf{v}_\varphi^{(t)}$.

**Primitive statement search**: Let $\mathbf{V}_\varphi = [\mathbf{v}_\varphi^{(1)}, ..., \mathbf{v}_\varphi^{(T)}]^\top$ be the output embedding of $T$ paths. The path attention is generated as

$$\mathbf{Q}_\psi = \hat{\mathbf{H}} + \mathbf{e}_\psi, \mathbf{Q}'_\psi = \hat{\mathbf{H}} + \mathbf{e}'_\psi, \qquad \tilde{\mathbf{V}}_\psi, \mathbf{S}_\psi = \mathrm{MultiHeadAttn}(\mathbf{Q}_\psi, \mathbf{V}_\varphi),$$
$$\tilde{\mathbf{V}}'_\psi, \mathbf{S}'_\psi = \mathrm{MultiHeadAttn}(\mathbf{Q}'_\psi, \mathbf{V}_\varphi), \qquad \mathbf{V}_\psi = \mathrm{FeedForward}(\mathrm{Concat}(\tilde{\mathbf{V}}_\psi, \tilde{\mathbf{V}}'_\psi)).$$

Here, $\mathbf{e}_\psi$ and $\mathbf{e}'_\psi$ are the first and second argument encodings, such that $\mathbf{Q}_\psi$ and $\mathbf{Q}'_\psi$ encode the arguments of each statement in $\Psi$. The compatibility between paths and the arguments are computed with two $\mathrm{MultiHeadAttn}$s. Finally, a $\mathrm{FeedForward}$ is used to aggregate the selections. Its output $\mathbf{V}_\psi \in \mathbb{R}^{K \times d}$ represents the results of all statement evaluations in $\Psi$.

**Formula search**: Let $\mathbf{Q}_{f,l}, \mathbf{Q}'_{f,l} \in \mathbb{R}^{C \times d}$ be the learnable queries of the first and second argument of formulas at $l$th level, and let $\mathbf{V}_{f,0} = \mathbf{V}_\psi$. The formula attention is generated as

$$\hat{\mathbf{V}}_{f,l-1} = [\mathbf{V}_{f,l-1} + \mathbf{e}_+, \mathbf{V}_{f,l-1} + \mathbf{e}_\neg], \qquad \tilde{\mathbf{V}}_{f,l}, \mathbf{S}_{f,l} = \mathrm{MultiHeadAttn}(\mathbf{Q}_{f,l}, \hat{\mathbf{V}}_{f,l-1}),$$
$$\tilde{\mathbf{V}}'_{f,l}, \mathbf{S}'_{f,l} = \mathrm{MultiHeadAttn}(\mathbf{Q}'_{f,l}, \hat{\mathbf{V}}_{f,l-1}), \quad \mathbf{V}_{f,l} = \mathrm{FeedForward}(\mathrm{Concat}(\tilde{\mathbf{V}}_{f,l}, \tilde{\mathbf{V}}'_{f,l})).$$

Here, $\mathbf{e}_+$, $\mathbf{e}_\neg$ are the learnable embeddings, such that $\hat{\mathbf{V}}_{f,l-1}$ represents the positive and negative states of the formulas at $l - 1$th level. Similar to the statement search, the compatibility between the logic `and` arguments and the previous formulas are computed with two $\mathrm{MultiHeadAttn}$s. And the embeddings of formulas at $l$th level $\mathbf{V}_{f,l}$ are aggregated by a $\mathrm{FeedForward}$. Finally, let $\mathbf{q}_o$ be the learnable final output query and let $\mathbf{V}_o = [\mathbf{V}_{f,0}, ..., \mathbf{V}_{f,L-1}]$. The output attention is computed as

$$\mathbf{v}_o, \mathbf{s}_o = \mathrm{MultiHeadAttn}(\mathbf{q}_o, \mathbf{V}_o).$$

## 5 STOCHASTIC TRAINING AND RULE VISUALIZATIONS

The training of NLIL consists of two phases: rule generation and rule evaluation. During generation, we run Eq.(15) to obtain the attentions $\mathbf{S}_\varphi, \mathbf{S}_\psi, \mathbf{S}'_\psi, \mathbf{S}_f, \mathbf{S}'_f$ and $\mathbf{s}_o$ for all $P^*$s. For the evaluation phase, we sample a mini-batch of queries $\{\langle \mathbf{x}, P^*, \mathbf{x}', y \rangle_i\}_{i=1}^b$, and evaluate the formulas using Eq.(14). Here, $y$ is the query label indicating if the triplet exists in the KB or not. We sample nonexistent queries to prevent the model from learning trivial rules that always output 1. In the experiments, these negative queries are sampled uniformly from the target query matrix $M^*$ where the entry is 0. Then the objective becomes

$$\arg\min_{\mathbf{w}} \frac{1}{b} \sum_i^b \mathrm{CrossEntropy}(y_i, \mathbf{s}_o^\top \mathbf{f}_L(\mathbf{x}, \mathbf{x}')).$$

Since the attentions are generated from Eq.(15) differentiably, the loss is back-propagated through the attentions into the Transformer networks for end-to-end training.

### 5.1 EXTRACTING EXPLICIT RULES

During training, the results from operator calls and logic combinations are averaged via attentions. For validation and testing, we evaluate the model with the explicit FOL rules extracted from the

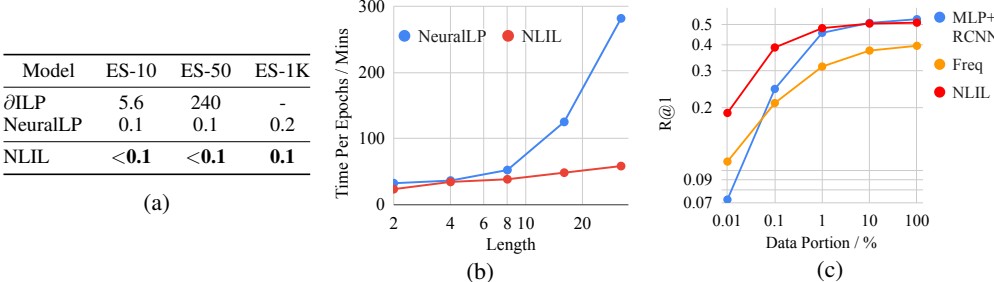

| Model | ES-10 | ES-50 | ES-1K |
|---|---|---|---|
| $\partial$ILP | 5.6 | 240 | - |
| NeuralLP | 0.1 | 0.1 | 0.2 |
| NLIL | **<0.1** | **<0.1** | **0.1** |

(a)                                (b)                                (c)

Figure 5: (a) Time (mins) for solving Even-and-Successor tasks. (-) indicates method runs out of time limit; (b) Running time for different rule lengths; (c) R@1 for object classification with different training set size.

attentions. To do this, one can view an attention vector as a categorical distribution. For example, $\mathbf{s}_\varphi^{(t)}$ is such a distribution over random variables $k \in [1, K]$. And the weighted sum is the expectation over $M_k$. Therefore, one can extract the explicit rules by sampling from the distributions (Kool et al., 2018; Yang et al., 2017).

However, since we are interested in the best rules and the attentions usually become highly concentrated on one entity after convergence. We replace the sampling with the $\arg\max$, where we get the one-hot encoding of the entity with the largest probability mass.

## 6  EXPERIMENTS

We first evaluate NLIL on classical ILP benchmarks and compare it with 4 state-of-the-art KB completion methods in terms of their accuracy and efficiency. Then we show NLIL is capable of learning FOL explanations for object classifiers on a large image dataset when scene-graphs are present. Though each scene-graph corresponds to a small KB, the total amount of the graphs makes it infeasible for all classical ILP methods. We show that NLIL can overcome it via efficient stochastic training. Our implementation is available at https://github.com/gblackout/NLIL.

### 6.1  CLASSICAL ILP BENCHMARKS

We evaluate NLIL together with two state-of-the-art differentiable ILP methods, i.e. NeuralLP (Yang et al., 2017) and $\partial$ILP (Evans & Grefenstette, 2018), and two structure embedding methods, TransE (Bordes et al., 2013) and RotatE (Sun et al., 2019). Detailed experiments setup is available at Appendix C.

**Benchmark datasets**: (i) Even-and-Successor (ES) benchmark is introduced in (Evans & Grefenstette, 2018), which involves two unary predicates `Even(X)`, `Zero(X)` and one binary predicate `Succ(X,Y)`. The goal is to learn FOL rules over a set of integers. The benchmark is evaluated with 10, 50 and 1K consecutive integers starting at 0; (ii) FB15K-237 is a subset of the Freebase knowledge base (Toutanova & Chen, 2015) containing general knowledge facts; (iii) WN18 (Bordes et al., 2013) is the subset of WordNet containing relations between words. Statistics of datasets are provided in Table 2.

**Knowledge base completion**: All models are evaluated on the KB completion task. The benchmark datasets are split into train/valid/test sets. The model is tasked to predict the probability of a fact triplet (query) being present in the KB. We use Mean Reciprocal Ranks (MRR) and Hits@10 for evaluation metrics (see Appendix C for details).

Results on Even-and-Successor benchmark are shown in Table 5a. Since the benchmark is noise-free, we only show the wall clock time for completely solving the task. As we have previously mentioned, the forward-chaining method, i.e. $\partial$ILP scales exponentially in the number of facts and quickly becomes infeasible for 1K entities. Thus, we skip its evaluation for other benchmarks.

Results on FB15K-237 and WN18 are shown in Table. 1. Compared to NeuralLP, NLIL yields slightly higher scores. This is due to the benchmarks favor symmetric/asymmetric relations or compositions of a few relations (Sun et al., 2019), such that most valuable rules will already lie within the chain-like search space of NeuralLP. Thus the improvements gained from a larger search space with NLIL are limited. On the other hand, with the Transformer block and smaller model created

for each target predicate, NLIL can achieve a similar score at least 3 times faster. Compared to the structure embedding methods, NLIL is significantly outperformed by the current state-of-the-art, i.e. RotatE, on FB15K. This is expected because NLIL searches over the symbolic space that is highly constrained. However, the learned rules are still reasonably predictive, as its performance is comparable to that of TransE.

**Scalability for long rules**: we demonstrate that NLIL can explore longer rules efficiently. We compare the wall clock time of NeuralLP and NLIL for performing one epoch of training against different maximum rule lengths. As shown in Figure 5b, NeuralLP searches over a chain-like rule space thus scales linearly with the length, while NLIL searches over a hierarchical space thus grows in log scale. The search time for length 32 in NLIL is similar to that for length 3 in NerualLP.

## 6.2 ILP ON VISUAL GENOME DATASET

The ability to perform ILP efficiently extends the applications of NLIL to beyond canonical KB completion. For example in visual object detection and relation learning, supervised models can learn to generate a *scene-graph* (As shown in Figure 1) for each image. It consists of nodes each labeled as an object class. And each pair of objects are connected with one type of relation. The scene-graph can then be represented as a relational KB where one can perform ILP. Learning the FOL rules on such an output of a supervised model is beneficial. As it provides an alternative way of interpreting model behaviors in terms of its relations with other classifiers that are consistent across the dataset.

To show this, we conduct experiments on Visual Genome dataset (Krishna et al., 2016). The original dataset is highly noisy (Zellers et al., 2018), so we use a pre-processed version available as the GQA dataset (Hudson & Manning, 2019). The scene-graphs are converted to a collection KBs, and its statistics are shown in Table 2. We filter out the predicates with less than 1500 occurrences. The processed KBs contain 213 predicates. Then we perform ILP on learning the explanations for the top 150 objects in the dataset.

Table 3: R@1 and R@5 for 150 objects classification on VG.

| Model | Visual Genome | |
|---|---|---|
| | R@1 | R@5 |
| MLP+RCNN | **0.53** | **0.81** |
| Freq | 0.40 | 0.44 |
| NLIL | 0.51 | 0.52 |

Quantitatively, we evaluate the learned rules on predicting the object class labels on a held-out set in terms of their R@1 and R@5. As none of the ILP works scale to this benchmark, we compare NLIL with two supervised baselines: (i) **MLP-RCNN**: a MLP classifier with RCNN features of the object (available in GQA dataset) as input; and (ii) **Freq**: a frequency-based baseline that predicts object label by looking at the mostly occurred object class in the relation that contains the target. This method is nontrivial. As noted in (Zellers et al., 2018), a large number of triples in Visual Genome are highly predictive by knowing only the relation type and either one of the objects or subjects.

**Explaining objects with rules**: Results are shown in Table 3. We see that the supervised method achieves the best scores, as it relies on highly informative visual features. On the other hand, NLIL achieves a comparable score on R@1 solely relying on KBs with sparse binary labels. We note that NLIL outperforms **Freq** significantly. This means the FOL rules learned by NLIL are beyond the superficial correlations exhibited by the dataset. We verify this finding by showing the rules for top objects in Table 4.

**Induction for few-shot learning**: Logic inductive learning is data-efficient and the learned rules are highly transferrable. To see this, we vary the size of the training set and compare the R@1 scores for 3 methods. As shown in Figure 5c, the NLIL maintains a similar R@1 score with less than 1% of the training set.

## 7 CONCLUSION

In this work, we propose Neural Logic Inductive Learning, a differentiable ILP framework that learns explanatory rules from data. We demonstrate that NLIL can scale to very large datasets while being able to search over complex and expressive rules. More importantly, we show that a scalable ILP method is effective in explaining decisions of supervised models, which provides an alternative perspective for inspecting the decision process of machine learning systems.

ACKNOWLEDGMENTS

This project is partially supported by DARPA ASED program under FA8650-18-2-7882. We thank Ramesh Arvind[3] and Hoon Na[4] for implementing the MLP baseline.

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

# A    RELATED WORK

Inductive Logic Programming (ILP) is the task that seeks to summarize the underlying patterns shared in the data and express it as a set of logic programs (or rule/formulae) (Lavrac & Dzeroski, 1994). Traditional ILP methods such as AMIE+ (Galárraga et al., 2015) and RLvLR (Omran et al., 2018) relies on explicit search-based method for rule mining with various pruning techniques. These works can scale up to very large knowledge bases. However, the algorithm complexity grows exponentially in the size of the variables and predicates involved. The acquired rules are often restricted to Horn clauses with a maximum length of less than 3, limiting the expressiveness of the rules. On the other hand, compared to the differentiable approach, traditional methods make use of hard matching and discrete logic for rule search, which lacks the tolerance for ambiguous and noisy data.

The state-of-the-art differentiable forward-chaining methods focus on rule learning on predefined templates (Evans & Grefenstette, 2018; Campero et al., 2018; Ho et al., 2018), typically in the form of a Horn clause with one head predicate and two body predicates with chain-like variables, i.e.

$$P^*(X, X') \leftarrow P_1(X, Y) \wedge P_2(Y, X').$$

To evaluate the rules, one starts with a *background* set of facts and repeatedly apply rules for every possible triple until no new facts can be deduced. Then the deduced facts are compared with a held-out ground-truth set. Rules that are learned in this approach are in first-order, i.e. data-independent and can be readily interpreted. However, the deducing phase can quickly become infeasible with a larger background set. Although $\partial$ILP (Evans & Grefenstette, 2018) has proposed to alleviate by performing only a fixed number of steps, works of this type could generally scale to KBs with less than 1K facts and 100 entities. On the other hand, differentiable backward-chaining methods such as NTP (Rocktäschel & Riedel, 2017) are more efficient in rule evaluation. In (Minervini et al., 2018), NTP 2.0 can scale to larges KBs such as WordNet. However, FOL rules are searched with templates, so the expressiveness is still limited.

Another differentiable ILP method, i.e. Neural Logic Machine (NLM), is proposed in (Dong et al., 2019), which learns to represent logic predicates with tensorized operations. NLM is capable of both deductive and inductive learning on predicates with unknown arity. However, as a forward-chaining method, it also suffers from the scalability issue as $\partial$ILP. It involves a permutation operation over the tensors when performing logic deductions, making it difficult to scale to real-world KBs. On the other hand, the inductive rules learned by NLM are encoded by the network parameters implicitly, so it does not support representing the rules with explicit predicate and logic variable symbols.

**Multi-hop reasoning**: Multi-hop reasoning methods (Guu et al., 2015; Lao & Cohen, 2010; Lin et al., 2015; Gardner & Mitchell, 2015; Das et al., 2016; Yang et al., 2017) such as NeuralLP (Yang et al., 2017) construct rule on-the-fly when given a specific query. It adopts a flexible ILP setting: instead of pre-defining templates, it assumes a chain-like Horn clause can be constructed to answer the query

$$P^*(X, X') \leftarrow P^{(1)}(X, Y_1) \wedge P^{(2)}(Y_1, Y_2) \wedge ... \wedge P^{(T)}(Y_{n-1}, X').$$

And each step of the reasoning in the chain can be efficiently represented by matrix multiplication. The resulting algorithm is highly scalable compared to the forward-chaining counter-parts and can learn rules on large datasets such as FreeBase. However, this approach reasons over a single chain-like path, and the path is sampled by performing random walks that are independent on the task context (Das et al., 2017), limiting the rule expressiveness. On the other hand, the FOL rule is generated while conditioning on the specific query, making it difficult to extract rules that are globally consistent.

**Link prediction with relational embeddings**: Besides multi-hop reasoning methods, a number of works are proposed for KB completion using learnable embeddings for KB relations. For example, In (Bordes et al., 2013; Sun et al., 2019; Balaževi\'c et al., 2019) it learns to map KB relations into vector space and predict links with scoring functions. NTN (Socher et al., 2013), on the other hand, parameterizes each relation into a neural network. In this approach, embeddings are used for predicting links directly, thus its prediction cannot be interpreted as explicit FOL rules. This is different from that in NLIL, where predicate embeddings are used for generating data-independent rules.

Table 4: Example rules learned by NLIL

| |
|---|
| $\mathtt{Person}(X) \leftarrow (\mathtt{Shirt}(Y_1) \wedge \mathtt{Wearing}(X,Y_1)) \vee (\mathtt{Pants}(Y_2) \wedge \mathtt{Wearing}(X,Y_2)) \vee$ $\quad\quad (\mathtt{Street}(Y_3) \wedge \mathtt{WalkingOn}(X,Y_3))$ |
| $\mathtt{Tree}(X) \leftarrow (\mathtt{Leaf}(Y_1) \wedge \mathtt{At}(Y_1,X)) \vee (\mathtt{SideWalk}(Y_2) \wedge \mathtt{Near}(Y_2,X))$ |
| $\mathtt{Shirt}(X) \leftarrow (\mathtt{Person}(Y_1) \wedge \mathtt{Wearing}(Y_1,X)) \vee (\mathtt{Child}(Y_2) \wedge \mathtt{Wearing}(Y_2,X))$ |
| $\mathtt{Sky}(X) \leftarrow (\mathtt{Clouds}(Y_1) \wedge \mathtt{in}(Y_1,X)) \vee (\mathtt{Airplane}(Y_2) \wedge \mathtt{Below}(X,Y_2))$ |
| $\mathtt{Head}(X) \leftarrow \mathtt{Helmet}(Y_1) \wedge \mathtt{Above}(Y_1,X)$ |
| $\mathtt{Head}(X) \leftarrow \mathtt{Wearing}(Y_1,Y_2) \wedge \mathtt{SittingOn}(X,Y_1) \wedge \mathtt{Hat}(Y_2)$ |
| $\mathtt{Sign}(X) \leftarrow (\mathtt{Number}(Y_1) \wedge \mathtt{On}(Y_1,X)) \vee (\mathtt{Post}(Y_2) \wedge \mathtt{On}(Y_2,X)) \vee$ $\quad\quad (\mathtt{Letter}(Y_3) \wedge \mathtt{In}(Y_3,X))$ |
| $\mathtt{Sign}(X) \leftarrow \mathtt{StreetLight}(Y_1) \wedge \mathtt{On}(Y_1,Y_2) \wedge \mathtt{On}(X,Y_2)$ |
| $\mathtt{Ground}(X) \leftarrow (\mathtt{Dog}(Y_1) \wedge \mathtt{On}(Y_1,X)) \vee (\mathtt{Grass}(Y_2) \wedge \mathtt{CoveredBy}(X,Y_2))$ |
| $\mathtt{Car}(X) \leftarrow \mathtt{Wheel}(Y_1) \wedge \mathtt{Of}(Y_1,X) \wedge \mathtt{Window}(Y_2) \wedge \mathtt{Of}(Y_2,X)$ |
| $\mathtt{Sidewalk}(X) \leftarrow \mathtt{Person}(Y_1) \wedge \mathtt{WalkingOn}(Y_1,X) \wedge \mathtt{Street}(Y_2) \wedge \mathtt{Near}(X,Y_2)$ |
| $\mathtt{Car}(X) \leftarrow \mathtt{Wheel}(Y_1) \wedge \mathtt{Of}(Y_1,X) \wedge \mathtt{Window}(Y_2) \wedge \mathtt{Of}(Y_2,X)$ |
| $\mathtt{Ear}(X) \leftarrow \mathtt{Eye}(Y_1) \wedge \mathtt{Of}(Y_1,Y_2) \wedge \mathtt{Of}(X,Y_2)$ |
| $\mathtt{Chair}(X) \leftarrow \mathtt{Arm}(Y_1) \wedge \mathtt{In}(Y_1,X) \wedge \mathtt{Person}(Y_2) \wedge \mathtt{SittingOn}(Y_2,X)$ |

## B   CHALLENGES IN ILP

Standard ILP approaches are difficult and involve several procedures that have been proved to be NP-hard. The complexity comes from 3 levels: first, the search space for a formula is vast. The body of the entailment can be arbitrarily long and the same predicate can appear multiple times with different variables, for example, the $\mathtt{Inside}$ predicate in Eq.(2) appears twice. Most ILP works constrain the logic entailment to be Horn clause, i.e. the body of the entailment is a flat conjunction over literals, and the length limited within 3 for large datasets.

Second, constructing formulas also involves assigning logic variables that are shared across different predicates, which we refer to as **variable binding**. For example, in Eq.(2), to express that a person is inside the car, we use $X$ and $Y$ to represent the region of a person and that of a car, and the same two variables are used in $\mathtt{Inside}$ to express their relations. Different bindings lead to different meanings. For a formula with $n$ arguments (Eq.(2) has 7), there are $\mathcal{O}(n^n)$ possible assignments. Existing ILP works either resort to constructing formula from pre-defined templates (Evans & Grefenstette, 2018; Campero et al., 2018) or from chain-like variable reference (Yang et al., 2017), limiting the expressiveness of the learned rules.

Finally, evaluating a formula candidate is expensive. A FOL rule is data-independent. To evaluate it, one needs to replace the variables with actual entities and compute its value. This is referred to as *grounding* or *instantiation*. Each variable used in a formula can be grounded independently, meaning a formula with $n$ variables can be instantiated into $\mathcal{O}(C^n)$ grounded formulas, where $C$ is the number of total entities. For example, Eq.(2) contains 3 logic variables: $X$, $Y$ and $Z$. To evaluate this formula, one needs to instantiate these variables into $C^3$ possible combinations, and check if the rule holds or not in each case. However in many domains, such as object detection, such grounding space is vast (e.g. all possible bounding boxes of an image) making the full evaluation infeasible. Many forward-chaining methods such as $\partial$ILP (Evans & Grefenstette, 2018) scales exponentially in the size of the grounding space, thus are limited to small scale datasets with less than 10 predicates and 1K entities.

## C   EXPERIMENTS

**Baselines**: For NeuralLP, we use the official implementation at **here**. For $\partial$ILP, we use the third-party implementation at **here**. For TransE, we use the implementation at **here**. For RotatE, we use the official implementation at **here**.

Table 5: Example low-accuracy rules learned by NLIL.

| |
|---|
| $\texttt{Bush}(X) \leftarrow \neg\texttt{Tree}(X)$ |
| $\texttt{Bus}(X) \leftarrow \neg(\texttt{Shirt}(Y_1) \wedge \texttt{Wearing}(Y_1, X))$ |
| $\texttt{Backpack}(X) \leftarrow \texttt{Person}(Y_1) \wedge \texttt{With}(Y_1, X)$ |
| $\texttt{Flowers}(X) \leftarrow \texttt{Pot}(Y_1) \wedge \texttt{With}(X, Y_1)$ |
| $\texttt{Dirt}(X) \leftarrow \texttt{Ground}(Y_1) \wedge \texttt{Near}(Y_1, X)$ |

**Model setting**: For NLIL, we create separate Transformer blocks for each target predicate. All experiments are conducted on a machine with i7-8700K, 32G RAM and one GTX1080ti. We use the embedding size $d = 32$. We use 3 layers of multi-head attentions for each Transformer network. The number of attention heads are set to number_of_heads $= 4$ for encoder, and the first two layers of the decoder. The last layer of the decoder has one attention head to produce the final attention required for rule evaluation.

For KB completion task, we set the number of operator calls $T = 2$ and formula combinations $L = 0$, as most of the relations in those benchmarks can be recovered by symmetric/asymmetric relations or compositions of a few relations (Sun et al., 2019). Thus complex formulas are not preferred. For FB15K-237, binary predicates are grouped hierarchically into domains. To avoid unnecessary search overhead, we use the most frequent 20 predicates that share the same root domain (e.g. "award", "location") with the head predicate for rule body construction, which is a similar treatment as in (Yang et al., 2017). For VG dataset, we set $T = 3$, $L = 2$ and $C = 4$.

**Evaluation metrics**: Following the conventions in (Yang et al., 2017; Bordes et al., 2013) we use Mean Reciprocal Ranks (MRR) and Hits@10 for evaluation metrics. For each query $\langle \mathbf{x}, P_k, \mathbf{x}' \rangle$, the model generates a ranking list over all possible groundings of predicate $P_k$, with other ground-truth triplets filtered out. Then MRR is the average of the reciprocal rank of the queries in their corresponding lists, and Hits@10 is the percentage of queries that are ranked within the top 10 in the list.

