# OpenReview forum: "Learn to Explain Efficiently via Neural Logic Inductive Learning"
_ICLR.cc/2020/Conference — Accept (Poster)_

### Official Review · AnonReviewer1 · 2019-10-17
**Official Blind Review #1**

**Rating:** 8

**Review:**

This paper proposes a novel architecture of integrating neural models with logic inference capabilities to achieve the goal of scalable predictions with explanatory decisions, which are of significant importance in the real deployment. In general, the article is well-written and nicely-structured with clear motivations, i.e., make the model predictions interpretable, make the logic rules differentiable, and make the algorithm scalable with longer rules and faster inference time. Both the methodology and experimental results make the idea promising and interesting. The contributions could be summarized in the following which make this piece of work worth of publication to my point of view:

1. I like the idea of introducing the operator concepts to save much time in the variable binding problems in the common logic inference domain.

2. The authors proposes a hierarchical attention mechanism to transform the rule composing problem into 3 stages with 3 different subspaces, namely operator composition, primitive statement composition and formula generation. This hierarchy structure solves the inductive problem efficiently by decomposing it into subproblems, which seems novel to me.

3. The proposal of a general logic operator defined in eq5 is crucial for formula generation in a differentiable way.

Despite the above contributions, there are a few limitations and questions to the author:

1. The paper states that eq(5) is able to represent various logic operators with proper parameters. Can you provide some examples of how this general formula represent simple operators such as "p \vee q"? It also mentions the case to avoid trivial expressions, but it's not clear how this is solved.

2. For operator search, I assume "e_X" indicates the representation for the head entity, then what does "e_Y" represent? If each operator at most takes the head entity as input, where does "e_Y" come from? does the process for operator search repeat for each different operator indicated by "e_\phi"? If this is the case, what's the effect of adding extra predicate embeddings "H_b"? Furthermore, the formula search is not clearly illustrated as of how eq(5) is softly picked using the defined process?

3. Section 4.2 introduces a use case for end-to-end evaluation through relational knowledge base.However, it is unclear to me how those score functions and "f_i^{(t)}" contribute to the search model, i.e., how those formulas in section 4.2 map back to the search functions introduced earlier? This is crucial to understand the gradient backpropagation. And it could be better to provide an illustrative algorithm on generating the actual rules from the search modules.

4. For the experimental section and related work, another existing work is missing, i.e., "Neural Logic Machines". More discussions and comparisons (experimental comparisons if possible) are helpful. A further question to ask is whether the proposed architecture could be used in the case when domain knowledge is not that explicit, e.g., the predicates are unknown or some of them are unknown?

**Experience Assessment:**

I have read many papers in this area.

**Review Assessment: Checking Correctness Of Derivations And Theory:**

I carefully checked the derivations and theory.

**Review Assessment: Checking Correctness Of Experiments:**

I carefully checked the experiments.

**Review Assessment: Thoroughness In Paper Reading:**

I read the paper thoroughly.

---

> ### Author Response · Authors · 2019-11-13
> **Response to Reviewer1**
>
> Thank you for your comments. Our responses to your questions are as follows.
>
>
> -How are formulas related to the search functions? How is gradient backpropagated?
>
> Our model is split into two parts: rule generation (Figure 4) and rule evaluation (Figure 3). In the rule generation phase, we run our Transformer model and generate a set of data-independent attentions. These attentions encode the soft picks of operators,  statements and the formulas.  After this phase,  the only things that go to the next phase are the attentions $\mathbf{S}_\varphi, \mathbf{S}_\psi, \mathbf{S}_\psi', \mathbf{S}_f, \mathbf{S}_f'$ and $\mathbf{s}_o$.
>
> In the rule evaluation (Figure 3), we use these attentions as weights to performs weighted sum over the matrices (eq10, eq13) which are fixed and never learned to generate the score and compare it to the groundtruth to generate gradient.  Since every step is differentiable, we can backprop the gradient through attentions into the Transformer module and the learnable embeddings.
>
>
>
> -What do "e_X" and e_Y (now e_{X'}) represent? what's the effect of adding predicate embeddings?
>
> During rule generation phase, we use Transformer to generate data-independent attentions by "mimicking" the evaluation happened in (Figure 3) with embeddings: $\mathbf{e}_X$ and $\mathbf{e}_Y$ (now $\mathbf{e}_{X'}$) are two "dummy" embeddings that mimics the query input, H is the embedding table of predicates and  encodings such as $\mathbf{e}_\varphi, \mathbf{e}_+, \mathbf{e}_\neg$ are used to alter the predicate and statements embeddings. This is similar to the sinusoidal encoding in the Transformer language tasks. Since all the computations in this phase are to "mimic" the actual evaluation, this part alone does not generate any query-specific outputs.
>
> Longer answer: We use the  $\mathbf{e}_X$ and $\mathbf{e}_{X'}$ embedding to replace the specific entity in the query so that we can generate rules that are data independent. And as described in section 4, we want to search body variables that are represented as the relational paths starting from the head variables. And for a binary target predicate, there are two "starting" points, i.e. $X$ and $X'$. So, indeed, each operator takes exactly one variable. What we want the model to learn is "which input to take". For example, for a target predicate $P^*(X,X')$ and two  operators $\varphi_1$ and $\varphi_2$, we explore the path $\{\varphi_1(X), \varphi_1(X'), \varphi_2(X), \varphi_2(X')\}$ at $(1)$th step, and $\{\varphi_1(\varphi_1(X)), \varphi_1(\varphi_1(X')), \varphi_1(\varphi_2(X)), \varphi_1(\varphi_2(X')), … \}$ at $(2)$th step and etc.
>
> Encoding such as $\mathbf{e}_\varphi$ are also used in this context. For rule generation phase (Figure 4), we generate the attentions between all possible inputs and all possible operators, which encode the soft choice of the relational paths. We use Transformer to compute the "compatibility" between the input value and the query, and we treat this compatibility as the attention we want. Here, the input and the queries are embeddings, for example, in the translation task, the inputs are word embeddings of the source sequence, and the queries are word embeddings of the target sequence. And the attention measures how likely each source word is associated with the target word.  In our case, the "sources" at $(0)$th step are the embeddings of $[\mathbf{e}_{X}, \mathbf{e}_{X'}]$ (and subsequently the outputs of previous Transformer module), the "targets" are the embeddings of all operators. Here, we have an embedding table H for all predicates but not for their "operators", so we make a learnable embedding $\mathbf{e}_\varphi$, essentially telling the model that "the embedding of an operator is the sum of its predicate embedding and $\mathbf{e}_\varphi$". This is similar to the sinusoidal encoding in the Transformer language tasks.

---

> ### Author Response · Authors · 2019-11-13
> **Response to Reviewer1 - cont'd**
>
> -Can you provide some examples of how this general formula represents simple operators such as "p \vee q"?
>
> (Details in sec 3.3) In our model, we consider soft "$\land$", and soft "$\neg$" logic operation because logic "$\vee$" is equivalent to $\neg(\neg p \land \neg q)$. Similar to the approach in path finding, we list out all the negations or a formula and the second level picks two to perform logic $\land$ . We take the example from sec 3.3 of the 2nd draft: if $\mathcal{F}_0 = \{f_1, f_2\}$ where each $f$ is a score $[0,1]$, then $\hat{\mathcal{F}}_0 = \{f_1,f_2,1-f_1,1-f_2\}$. With $C=2$, the next level $\mathcal{F}_1$ could become $\{(1-f_1)*(1-f_2), f_1*(1-f_2)\}$, by augmenting it with its negation we have  $\hat{\mathcal{F}}_1 = \{(1-f_1)*(1-f_2), f_1*(1-f_2), (1-(1-f_1)*(1-f_2)), (1-f_1*(1-f_2))\}$, so here $(1-(1-f_1)*(1-f_2))$ is the soft logic or of the two original operands.
>
>
>
>
> -It also mentions the case to avoid trivial expressions, but it's not clear how this is solved.
>
> The trivial expressions such as $(q \land \neg q)$ will result in a formula that always outputs 1 or 0. This can be a problem if we only train our model with positive queries, i.e. with the label of 1. So we solve this by introducing negative queries during training (now described in sec 5), such that these trivial rules will be penalized.
>
>
>
> -The formula search is not illustrated as of how it is softly picked
>
> We apologize for the confusion. In our revision, we made it clear in section 3 that, the objective of our model is to learn a set of attentions $\mathbf{S}_\varphi, \mathbf{S}_\psi, \mathbf{S}_\psi', \mathbf{S}_f, \mathbf{S}_f'$ and $\mathbf{s}_o$ that describe the soft picks of operators, statements, and formulas respective (Figure 4). During training, we use attentions to perform weighted sum over the matrices to get the output score (Figure 3). For validation, testing and visualization (as stated in sec 5), we simply take the argmax over these attentions, and that gives us the hard samples that the model picks. For example, for a single input $\mathbf{e}_X$ and two operators $[\varphi_1, \varphi_2]$ we have the attentions of shape 2X1, say it's $[0.8, 0.2]$. During training, we perform weighted sum $(0.8\mathbf{M}_1 + 0.2\mathbf{M}_2)\mathbf{v}_\mathbf{x}$ as shown in eq5. And for rule extraction, we take argmax and get $[1, 0]$, so we visualize the rule as "Phi_1(X)"
>
>
>
> -Discussions and comparison with Neural Logic Machines is missing
>
> We thank the reviewer for noticing this related work. Yes, we are aware of the NLM model. However, we didn't include this in our initially draft for two reasons: (i) NLM has similar scalability as diff-ILP. We have conducted a few small-scale experiments that have a similar scale as the Even-Succ benchmark using the official NLM implementation, and found that it generally took it more than 30 minutes to solve in our environment. So it doesn't scale to KBs such as FB15K. As stated in the NLM paper and rebuttal (https://openreview.net/forum?id=B1xY-hRctX ), NLM is not very efficient at handling "both complex reasoning rules and large-scale entity sets" which we consider are the two main motivations of our work.
>
> And most importantly, (ii) the NLM model does not learn to represent the FOL rule explicitly. NLM parameterizes logic operations into a sequence of MLPs. And as stated in their rebuttal that "NLM models do not explicitly encode FOPC logic forms". One of the main motivations of our work is to learn explanations that can be explicitly interpreted as FOL rules. So when designing our experiments, we mainly planned to compare with those rule learning models such as diff-ILP and NeuralLP, plus some supervised models, i.e. TransE and MLP+RCNN, that show the state-of-the-art performance on accuracy.
>
> However, we do agree with the reviewers that this should be stated clearly in the paper. We have included a discussion on NLM in our 2nd draft.
>
>
>
> -Whether the proposed architecture could handle unknown or ambiguous predicates?
>
> Thank you for the suggestion. In our 2nd draft, we have made an explicit connection to the multi-hop reasoning literature. Our model generally falls into this category as well but with some ILP-specific extensions. This approach assumes that it's operating on a fixed KB where all relations and facts are given. So yes, the model currently does not support the learning from unknown or ambiguous predicates. But we do agree this is an important direction to explore in the future.

---

### Official Review · AnonReviewer2 · 2019-10-23
**Official Blind Review #2**

**Rating:** 6

**Review:**

This paper presents a model for effectively hierarchically ‘searching’ through the space of (continuously relaxed) FOL formulas that explain the underlying dataset. The model presented employs a three-level architecture to produce logic entailment formulas, skolemized with a set of skolem functions, i.e. ‘operators’. The three-level search, which is implemented via a stack of transformers, first searches through a space of variable and predicate embeddings to select operators, after which it searches through the space of predicates to form primitive statements of predicates and operators, and finally it generates a number of formulas from the previously ‘selected’ primitive statements. The model is applied on a toy task to showcase its speed, a knowledge base completion task, and modeling the visual genome dataset, where the model shows the ability to induce meaningful rules that operate on visual inputs. The presented benefit of the model is scalability, ability to induce explainable rules, and the ability to induce full FOL rules.

The paper is well motivated from the explainability perspective and based on the evaluation does what it claims. The model is fairly elaborate, yet manages to be faster than the competing models. In general, I think the model itself is a welcome addition to the area of neuro-symbolic models, especially logic-inducing models, and that the evaluation done is appropriate (with a few caveats). However, my major critique of the paper is in its clarity.
In the current state, the paper is quite difficult to read, partially due to its density, partially due to its confusing presentation, notational issues and missing details. It would be difficult to reimplement the model just by reading the paper, which brings me to ask: will you be releasing the code?
I would be willing to accept the paper if the authors improve the notation and presentation significantly. I’m enumerating issues that I found most confusing:

Result presentation:
- The figure captions are uninformative. In figure 1, one needs to understand what the graph is before reading the text at the end of the paper which explains what that is. It is not clear from the figure itself. Figure 2 presents the model, but it does not follow the notation from the main body of the paper.
- Table 1 is missing SOTA models. TransE is definitely not one of the better models out there: check [2, 3, 4] for SOTA results which are significantly higher than the ones presented. I would not at all say that that invalidates the contribution of the paper, but readers should have a clear idea of how your model ranks against the top performing ones.
- Please provide solving times for TransE, as it has to be by far the fastest method, given that it is super-simple.
- Are provided times training or inference times (in each of the table/figure) because one gets mixed statements from the text?
- In which units is time in Table 1?
- Can you include partially correct or incorrect learned rules in Table 4? It would be great to get some understanding of what can go wrong, and if it does, what might be the cause.

Model presentation and notation:
- You mention negative sampling earlier in the text but then don’t mention how you do it
- Notation below (6) is utterly confusing and lacking: what is s_{l,i}^(t), is there a softmax somewhere, what are numbers below ‘score’
- What is the meaning of e_+ and e_- given that you omit details of negative sampling?
- The notation does not differentiate between previous modules well so there’s V^(t-1) across modules, and it is not clear which one is used at the end --- last choice over V^(0) - V^(T) is over the LAST output, not the output from previous steps?
- The notation in the text does not follow the notation in the figure (V_ps, V_\phi)
- Notation gets quite messy at some points, e.g. R^c^(0), is e_X and e_Y in H or not? Is H_b there too?
- The differentiation of embedding symbols is not done well. H_b is an embedding for binary predicates, or a set of predicates? Does that mean that there is only a single embedding for a binary predicate and a single embedding for its operator (thought I thought that operators have an embedding, each)?
- The explanation of what a transformer does is not particularly useful, the paper would benefit more from an intuitive explanation, such as that the transformer learns the ‘compatibility’ of predicates (query) and input variables (values), etc.
- The ‘Formula generation’ subsection lacks the feel of fitting where it is right now, given that the notation in it is useful only in ‘Formula search’ paragraph. The other thing is that that subsection is wholly unclear: where do p and q come from, do they represent probabilities of predicates P and Q? How are they calculated? Does your construction imply that a search over the (p, q, a) space is sufficient to cover all possibilities of and/not combinations? In which case alpha is a vector of multiple values different for each f_s  in (5) or no? It is unclear
- What is the relationship between alpha_0 and alpha_1, sum(alpha_0, alpha_1) = 1? Are alpha_0, and alpha_1 scalars, and how are they propagated in eq 5? Because if they’re just scalars, the continuous relaxation of formulas represented in (5) cannot cover all combinations of p, q, not and and. What is the shape of the alpha vector?
p and q are probabilities of what, predicates P and Q? How are they produced?
- are \phi functions pretrained?

Related work:
- I do not condone putting it into the appendix, but I’m not taking it as a make-or-break issue.
- It is notably missing an overview of symbolic ILP, and a short spiel on link prediction models (given that you compare your model to TransE, as a representative of these models)

The paper is riddled with typos and consistent grammatical errors. I would suggest re-checking it for errors. Examples include:
- singular/plural noun-verb mismatches (rules..and does not change -> do, Methods ...achieves -> achieve)
- Q_r^(t) - choice of left operands -> right
- An one-hot -> a one-hot


Minor issues:
- The claim that the NeuralLP is the only model able to scale to large datasets is a bit too strong given [1]
- You say “We could adopt” but you “do adopt”

Clarification questions:
- Are 10 times longer rules of any use? Can you provide an example of such rules, a correct one and an incorrect one?
- How many parameters does your model use? What are the hyperparameters used in training?
- How big is T in your experiments, and why?
- Why are queries only binary predicates?
- How discrete do these rules get? You sample them during testing and evaluation, but are embeddings which encode them approximately one-hot, so that the (Gumbel-softmax) sampling produces the same rules all over again or are these rules fairly continuous and sampling is just a way to visualise them, but they are internally much more continuous?
- Just to confirm, \phi are learned, right? Given that they are parameterised with a predicate matrix, and that matrix is trainable?
- Do you have a softmax on the output? It seems  f^*(T+1) should be a softmaxed value?
- The rule generation generates a substantial number of rules, right? What might be the number of these rules? Does the evaluation propagate gradients through all of them or to a top-k of them?
- Why is the formula in Eq.4 written the way it is. I assume it can be written differently, for example “Inside(\phi_Identity(X), \phi_Car())) and Inside(\phi_Clothing(), \phi_Identity(X))”. I do not understand why Clothing was treated as an operator and Car as a predicate, while treating Inside both as an operator and a predicate. Sure, nothing in the model forces it to consistently represent a formula in the same way always, but an example such as this one would need a good explanation why you chose it or at least mention that this is but one way to present it.
- Why is Identity(X) used? Is it because you did not want to mix in variable embeddings during the primitive statement search?


[1] Towards Neural Theorem Proving at Scale
[2] Canonical Tensor Decomposition for Knowledge Base Completion
[3] RotatE: Knowledge Graph Embedding by Relational Rotation in Complex Space
[4] TuckER: Tensor Factorization for Knowledge Graph Completion

**Experience Assessment:**

I have published one or two papers in this area.

**Review Assessment: Checking Correctness Of Derivations And Theory:**

I assessed the sensibility of the derivations and theory.

**Review Assessment: Checking Correctness Of Experiments:**

I assessed the sensibility of the experiments.

**Review Assessment: Thoroughness In Paper Reading:**

I read the paper at least twice and used my best judgement in assessing the paper.

---

> ### Author Response · Authors · 2019-11-13
> **Response to Reviewer2**
>
> Thank you for your comments. Our responses to your questions are as follows.
>
>
> -The paper is quite difficult to read
>
> We apologize for the writing in our initial draft. We have made significant modifications in our 2nd draft mainly in section 2, 3, 4 and 5. Specifically, we draw a close connection between our model with the existing literature of  multi-hop reasoning methods, and have made our notations and presentations align with the literature. We hope this will make more sense to the readers.
>
>
> -Will you be releasing the code?
>
> Yes, we plan to release the code. We have uploaded the main part of our implementation into the public repository https://github.com/gblackoutwas4/NLIL . Currently, we are still in the process of cleaning up the code, so there can be glitches in running the model. Hopefully, we will release the complete version in the next few weeks.
>
>
> -Overview of symbolic ILP and link prediction models
>
> We thank the reviewer for the suggestions on the related work section. We have included the review on traditional ILP, link prediction models and the recent Neural Logic Machine model.
>
>
>
> -Why is the formula in Eq.7 (eq4 in the initial draft) written the way it is? Can it be written differently?
>
> Yes, there are multiple ways in converting it into the operator form. Thanks for pointing this out. We have put an explanation right below this example.
>
>
>
> -Are \phi functions pretrained?
>
> Our model does not learn the logic operator itself, i.e. $\varphi$. It learns the weights that combine the operators for producing the results. We apologize for the confusion here. We have made significant modifications for this part (sec 3,4 and Figure 3,4) in our 2nd draft.
>
> Our model closely follows the setup in the multi-hop reasoning literature. It operates over a KB that contains a fixed set of predicates, where each of them is represented as a binary matrix. These matrices are fixed and never learned, because they encode the KB itself. The goal of the model is, when given a query $\langle x, P^*, x' \rangle$, to find a relational path that lead from $x$ to $x'$, where the path itself is parameterized as the chain of matrix multiplications on the one-hot vector of the input (as shown in eq3 of the 2nd draft). Intuitively, the model learns what matrix to multiply at each step that could lead to the correct results.
>
> In sec 3, we discuss how these hard-choices of picking matrices can be relaxed into taking the weighted sum over the matrices and over the scores (eq10,13 and Figure 3). So what our model learns is a set of attentions that guide the weighed sums. Therefore, even though the logical operators are parameterized into adjacency matrices, they are not learned by the model.
>
>
>
> -Are embeddings which encode the rules approximately one-hot? How to get discrete rules?
>
> Embeddings are used in rule generation (Figure 4) to generate data-independent attentions by "mimicking" the evaluation computations (Figure 3). Thus, embeddings themselves are not one-hot, it's the generated attentions that are representing the soft choices and can become one-hot with sampling.
>
> Longer answer: Our model is split into two parts: rule generation (Figure 4) and rule evaluation (Figure 3). In the rule generation phase, we run our Transformer model and generate a set of attentions (described in sec 4). This encodes the soft picks of operators,  statements and the formulas. And this part alone does not generate any query-specific outputs. It only operates over a set of embeddings: embeddings of "dummy" input $\mathbf{e}_X$, $\mathbf{e}_{X'}$, and embeddings of predicates $\mathbf{H}$. All the computations in this phase are to "mimic" the actual evaluation that could happen when the true query is fed (which will happen in the next phase), and the goal is to generate the attentions along this simulated evaluation. After this phase,  the only things that go to the next phase are the attentions $\mathbf{S}_\varphi, \mathbf{S}_\psi, \mathbf{S}_\psi', \mathbf{S}_f, \mathbf{S}_f'$ and $\mathbf{s}_o$.
>
> During training, we use attentions to perform weighted sums over the matrices to get the output score (eq10,13, Figure 3). For validation, testing and visualization (discussed in sec 5), we simply take the argmax over these attentions, and that gives us the one-hot hard samples that the model picks (originally, we were doing gumbel-softmax to sample from attentions, but it doesn't quite make sense, because during testing, we only concern the best rule learned by the model). For example, for a single input $\mathbf{e}_X$ and two operators $[\varphi_1, \varphi_2]$. For one step of operator call search, we will obtain the attentions of shape 2X1, say it's $[0.8, 0.2]$. During training, we perform weighted sum $(0.8\mathbf{M}_1 + 0.2\mathbf{M}_2)\mathbf{v}_\mathbf{x}$ as shown in eq5. And for rule extraction, we take argmax and get $[1, 0]$, so we visualize the rule as "Phi_1(X)"

---

> ### Author Response · Authors · 2019-11-13
> **Response to Reviewer2 - cont'd**
>
> -Do you have a softmax on the output? It seems  f^*(T+1) should be a softmaxed value?
>
> Yes, sorry for the confusion. The output of each statement is an unnormalized scalar score and is sigmoided into $[0, 1]$. This is now shown in eq9 and eq10. The following logic combinations are either (1 - score) or (score * score). And the weighted sums over them are also normalized, so it is still in $[0, 1]$.
>
>
>
> -The rule generation generates a substantial number of rules, right? What might be the number of these rules?
>
> Thank you for pointing this out.  We have updated the formula generation at section 3.3. Yes, we have a memory limit of $C$ for each level of logic combinations. For example, in $(l-1)$level we have 2 formulas $\{f_1, f_2\}$. We augment it with its negation, so there are 4 $\{f_1, f_2, 1-f_1,1-f_2\}$. Then all possible logic and combinations are $\{f_1*f_2, f_1*(1-f_1), f_1*(1-f_2), f_2*(1-f_1), f_2*(1-f_2), (1-f_1)*(1-f_2)\}$. So if $C=2$, we only keep 2 at the $(l)$level set. For differentiable training. this selection is parameterized as the weighted sums over the candidates $\{f_1,f_2,1-f_1,1-f_2\}$ w.r.t the two formula attention matrices. In this example, we have 4 candidates and 2 combinations, each combination has left and right operands, so the $(l)$level matrices have the shape of (2,4) and (2,4), one selecting the left operand and the other selecting the right operand.
>
>
> -What is the meaning of e_+ and e_- given that you omit details of negative sampling?
>
> $\mathbf{e}_\varphi, \mathbf{e}_+, \mathbf{e}_\neg$ are encodings used in rule generation, so they are not related to the negative sampling. During rule generation phase, we use Transformer to generate data-independent attentions by "mimicking" the evaluation happened in (Figure 3) with embeddings. Encodings such as $\mathbf{e}_\varphi, \mathbf{e}_+, \mathbf{e}_\neg$ are used to change the predicate and statements embeddings. This is similar to the sinusoidal encoding in the Transformer language tasks.
>
> Longer answer: In rule generation, we generate the attentions between all possible inputs and all possible operators, which encode the soft choice of the relational paths. To compute this, we use Transformer to compute the "compatibility" between the input value and the query, and we treat this compatibility as the attention we want. Here, the inputs and the queries are both embeddings, for example, in the translation task, the inputs are word embeddings of the source sequence, and the queries are word embeddings of the target sequence. And the attention measures how likely each source word is associated with the target word.  In our case, the "sources" at $(l)$th level of logic combination are the embeddings of formulas from $(l-1)$th level $\mathbf{V}_{f,l-1}$ (where $\mathbf{V}_{f,0}=\mathbf{V}_\psi=[\mathbf{v}_{\psi,1}, …, \mathbf{v}_{\psi,K}]$), the "targets" are two learnable formula queries of shape $(C, d)$. We want to compute the compatibility of each formula query with the ones from $(l-1)$th level and their negations (As stated in section 3.3, the formula set will be augmented with its negation). Here, we have the embedding $\mathbf{V}_{f,l-1}$, so we make two learnable embeddings $\mathbf{e}_+, \mathbf{e}_\neg$, essentially telling the model that "the embedding of a positive formula is the sum of its embedding and $\mathbf{e}_+$" and "the embedding of a negative formula is the sum of its embedding and $\mathbf{e}_\neg$". This is similar to the sinusoidal encoding in the Transformer language tasks.
>
>
>
> -What is H_b? Is there a single embedding for a binary predicate and a single embedding for its operator?
>
> $H_b$ is the stack of embeddings of unary predicates. This is actually the notion only to be used in the implementation and is not necessarily needed in the paper. In our 2nd draft, we have simplified our model, so this notation is no longer needed. In general, for $K$ predicates we learn an embedding table of shape $(K, d)$ where each row is the embedding of a particular predicate. Similar to the above, we do not learn the operator embedding individually, instead, we learn an operator encoding $\mathbf{e}_\varphi$, such that $\mathbf{h}_k + \mathbf{e}_\varphi$ produces the operator embedding.
>
>
>
> -Not mentioning negative sampling
>
> We have now included the description of negative sampling in sec 5. When we sample positive queries, for example, $\langle x, P^*, x' \rangle$, we also uniformly sample from the matrix of the target predicate $\mathbf{M}^*$ where the entry is 0, which is the negative samples.
>
>
>
> -Experiments on TransE and RotatE
>
> We have included the performance and time for TransE and RotatE in the 2nd draft.

---

> ### Author Response · Authors · 2019-11-13
> **Response to Reviewer2 - cont'd 2**
>
> -Are 10 times longer rules of any use? Can you provide examples?
>
> In our experiments, we find that longer rules are not effective in KB completion task on FB15K and WN18, because these two "benchmarks favor symmetric/asymmetric relations or compositions of a few relations". So the time vs length experiment shown in Figure 4(b) is more about illustrating our model's scalability rather than showing its expressiveness.  But in VG dataset, we do find the longer rules are more effective.  For example, the first rule in Tab 4 is the disjunction of 3 statements, which is more expressive than Horn clause rules. As requested, we have shown some incorrect rules in the Appendix Tab 5 as well.
>
>
>
> -What are the hyperparameters used in training? How big is T?
>
> We have updated the experiment detail section in Appendix C to report the hyperparameters we used. Generally, we use a latent dimension of 32, for each Transformer we use the layer size of 3 and 4 attention heads per layer.  For KB completion, we set $T$ to be 2 and zero logic combination, because all rules that can be learned from these two benchmarks are chain-like and short. For VG, we set $T$ to be 3 and include 2 levels of logic combinations each with $C=4$.
>
>
> -Why are queries only binary predicates?
>
> In this work, we assume all predicates are unary or binary, partially due to that the literature and baselines generally have the same assumption, and partially due to that most available large-scale KBs only come with unary attributes and binary relations.
>
>
> -Why is Identity(X) used?
>
> In the initial draft, we introduced this "imaginary" operator just for notation convenience, such that every variable in the body is now the result of the operator calls. They aren't necessarily needed for the model to run properly. We have removed this notion in the 2nd draft. Apologies for the confusion here.

---

> > ### Comment · AnonReviewer2 · 2019-11-14
> > **Reply + update**
> >
> > Having read the detailed comments and the updated paper, I’m happy with increasing my score. All the concerns I raised were well addressed and incorporated in the paper. However, the addition of a thorough focus on multi-hop reasoning and significant expansion of chapter three push this paper to the limit of 10 pages and since the reviewer instructions say: “Reviewers will be instructed to apply a higher standard to papers in excess of 8 pages.”, I’m hesitant to further increase the score of the paper.
> >
> > I do have another question, though: How come TransE model is slower than your model given that the score function for that model is extremely simple and easy to trivially batch and run even on large batches (several thousands)?

---

> > > ### Author Response · Authors · 2019-11-14
> > > **Response**
> > >
> > > We greatly appreciate the reviewer's reassessment on the paper.
> > >
> > > Regarding the reviewer's question. This is because we have a special optimization in our implementation. In the KB experiments, we have separate Transformer blocks for each target predicate (Appendix C), so the rule generation for each target is independent. Therefore, when looping through all the queries, we have a probability to skip the query if the rules for the target predicate associated with this query already have a high score on the validation set. So for a KB where relations are few and rules are highly predictive, such as WN, once we learn the asymmetric/symmetric rules, we can skip a lot of the queries, so we are faster than TransE. But for FB15K where there are many relations and rules are less predictive, the gain from this optimization is smaller, so we are slower than TransE.

---

> ### Comment · AnonReviewer2 · 2019-11-14
> **Forgotten reference**
>
> I just noticed I forgot to mention the following reference: Towards Neural Theorem Proving at Scale (https://arxiv.org/abs/1807.08204) . In the text you mention(ed) that NeuralLP is the 'only' 'scalable' method. This paper shows a way how to scale Neural Theorem Provers to WN and FB datasets and should be referenced.

---

> > ### Author Response · Authors · 2019-11-14
> > **Reference Added**
> >
> > We thank the reviewer for suggesting this related work. We have now included it in our related work section.

---

### Official Review · AnonReviewer3 · 2019-10-23
**Official Blind Review #3**

**Rating:** 3

**Review:**

The paper proposes to determine explanations for predictions using first-order logic (FOL). This requires being able to learn FOL rules. The authors propose to divide the search for FOL rules into 3 parts, each being hierarchical. The first level is a search for operator, followed by primitive statement search, followed by search for the Boolean formula. The authors also propose to implement logical variables that appear only in the body of the rule (and thus are existentially quantified) using skolem functions which reduces to a search operation and is an interesting idea which I haven't seen in recent works combining logic and neural networks. The paper then proposes a parameterized logical operator and describes their architecture for training these using attention and transformers.

I found the paper to be very difficult to read. For instance,  Equation (5) doesn't mention parameters \alpha on the RHS. I can't make out what the parameterization of the logical operator is. I can't also connect section 4 to the parameterized logical operator in section 3. Then Section 4.2 presents a score formulation (Equation 6) and refers the reader to the NeuralLP paper. This is not my favorite way to write a paper. So many indirections make it very difficult to appreciate the contributions. I hope the authors take this feedback and try re-writing their paper with the reader in mind.

Is there a specific reason why Neural Logic Machines (Dong et al, ICLR 2019) is not referenced? It also claims to be more efficient than \partial-ILP and to be able to learn rules. This is an important question. From what I can make out, not only should this paper cite Neural Logic Machines but they should in fact be comparing against it via experiments. It would also be helpful if the author present experiments against ILP systems (e.g. AMIE+). While ILP cannot deal with noisy labels, these full fledged systems do have some bells and whistles and it would be interesting to find out exactly what results they return. I couldn't make out exactly what the authors meant with this statement from the Appendix:
"The main drawback of NeuralLP is that the rule generation dependents on the specific query, i.e. it’s data-dependent. Thus making it difficult to extract FOL rules ..."
NeuralLP does learn FOL rules (of a particular form). I don't understand what the above statement means. I think the authors need to reference NeuralLP more carefully lest their statements come off as being too strong.

Two questions requiring further clarification:

- Since your logical operator is parameterized, how do you take a learned operator and identify which logical operator it corresponds to? More generally, how do you derive the crisp rules shown in Table 4 in the appendix?

- Since your network is fairly deep (e.g., I don't see a direct edge from the output layer to the operator learning layers in Fig 2), how do you ensure that gradients do not vanish? For instance, Neural Logic Machines use residual connections to (partially) address this. Is this not a problem for you?


**Experience Assessment:**

I have published one or two papers in this area.

**Review Assessment: Checking Correctness Of Derivations And Theory:**

I assessed the sensibility of the derivations and theory.

**Review Assessment: Checking Correctness Of Experiments:**

I assessed the sensibility of the experiments.

**Review Assessment: Thoroughness In Paper Reading:**

I read the paper at least twice and used my best judgement in assessing the paper.

---

> ### Author Response · Authors · 2019-11-13
> **Response to Reviewer3**
>
> Thank you for your comments. Our responses to your questions are as follows.
>
>
>
> -I found the paper to be very difficult to read.
>
> We apologize for the writing in our initial draft. We have made significant modifications in our 2nd draft mainly in section 2, 3, 4 and 5. Specifically, we draw a close connection between our model with the existing literature of  multi-hop reasoning methods, and have made our notations and presentations align with the literature. We hope this will make more sense to the readers.
>
>
>
> -Why Neural Logic Machines is not referenced and compared?
>
> We thank the reviewer for noticing this related work. Yes, we are aware of the NLM model. However, we didn't include this in our initially draft for two reasons: (i) NLM has similar scalability as diff-ILP. We have conducted a few small-scale experiments that have a similar scale as the Even-Succ benchmark using the official NLM implementation, and found that it generally took it more than 30 minutes to solve in our environment. So it doesn't scale to KBs such as FB15K. As stated in the NLM paper and rebuttal (https://openreview.net/forum?id=B1xY-hRctX ), NLM is not very efficient at handling "both complex reasoning rules and large-scale entity sets" which we consider are the two main motivations of our work.
>
> And most importantly, (ii) the NLM model does not learn to represent the FOL rule explicitly. NLM parameterizes logic operations into a sequence of MLPs. And as stated in their rebuttal that "NLM models do not explicitly encode FOPC logic forms". One of the main motivations of our work is to learn explanations that can be explicitly interpreted as FOL rules. So when designing our experiments, we mainly planned to compare with those rule learning models such as diff-ILP and NeuralLP, plus some supervised models, i.e. TransE and MLP+RCNN, that show the state-of-the-art performance on accuracy.
>
> However, we do agree with the reviewers that this should be stated clearly in the paper. We have included a discussion on NLM in our 2nd draft.
>
>
>
> -Experiments against ILP systems (e.g. AMIE+)
>
> Thank you for the suggestion. We didn't include traditional ILP methods such that AMIE+ into our experiments because it seems rather uncommon to take it as a baseline in many recent differentiable ILP works such as diff-ILP, NTP and NeuralLP. But we do agree with the reviewer that such comparison would be very helpful. We are currently exploring AMIE+ with our benchmarks. Given the relatively short rebuttal period, we may not be able to report the results before the deadline, but we do plan to include it in our later draft of the paper.
>
>
>
>
> -What does it mean "NeuralLP is data-dependent"?
>
> We apologize for the confusion here.  We have revised this argument in our paper and presented it formally in section 2.2. The NeuralLP is closely related to the multi-hop reasoning methods on KB, whereupon given a query, say $\langle x_1, P^*, x_1' \rangle$, the model searches for a relational path that leads from $x_1$ to $x_1'$. The relational path can then be interpreted in the ILP context as a chain-like FOL rule. However, when generating the path, the model is typically conditioned on the specific query objects (eq.5), i.e. $x_1$ and $x_1'$. Thus for a different query with the same predicate type, say $\langle x_2, P^*, x_2' \rangle$, the generated path can be different. This is what we refer to as "data-dependent", because the rule is local and does not guarantee to generalize to other instances.
>
> As one of the main motivations of our work, we'd like to learn the explanations that describe the "lifted" knowledge behind the decisions, so we think to be able to learn rules that are guaranteed to generalize to all instances is an important aspect of the proposed model. In other words, for all possible queries $\langle x_i, P^*, x_i' \rangle$ associated with target predicate $P^*$, the rule generated by our model should remain unchanged (eq.15).
>
> We agree with the reviewers that this consideration wasn't made very clear in the initial submission. In our revision, we have made this conditioning explicit with formal notations and equations.  Hopefully, this could make better sense to the readers.

---

> > ### Comment · AnonReviewer3 · 2019-11-14
> > **Still unclear about the basics despite the (significant) rewrite**
> >
> > I appreciate the effort put into the updated draft. Its a significant improvement over the previous version. However, I am still confused about the basics. Here are a couple of pointed questions:
> >
> > - In section 2.1 last para, you state that you consider both unary and binary predicates. Then why is your KB of the form {<x, P, x'>} ? If P is unary then what does the second constant refer to? Are you repeating the first constant?
> >
> > - In section 2.2, you state that every predicate is stored as a square, binary matrix. What does this mean if the predicate is unary? Do you end up utilizing just the diagonal of the matrix?
> >
> > - In equation (4), what is K? Is it the number of predicates in your KB? Where is this stated?
> >
> > - Also in equation (4), is $s_{\varphi,k}^{(t)}$ a scalar? In the same equation, is $s_{\psi}^{(t')}$ also a scalar?
> >
> > - In equation (3), I understood the superscript T on top of $v_{x'}$ to mean the transpose operator. However, you have another T on top of the product operator in the very same equation which denotes the length of the path, is that right? If so, is this overuse of the same letter to denote two very different things really necessary?
> >
> > - Where you define $s_{\psi}$ just after equation 4, what do the two T's mean?

---

> > > ### Author Response · Authors · 2019-11-14
> > > **Response**
> > >
> > > We thank the reivewer3 for reassessing the paper. Our responses to your questions are as follows.
> > >
> > >
> > > - What is {<x, P, x'>} for unary P?
> > >
> > > If the query predicate is unary then there is only one entity. For notation consistency, we use the same format. One can treat the second argument of the unary query as a placeholder or a copy of the first argument.
> > >
> > >
> > >
> > > - What is the matrix for unary $P_k$? Is it diagonal?
> > >
> > > Usually, in the multi-hop reasoning literature, people only consider binary predicates. So in section 3.1, we introduce how to incorporate unary predicates as matrices. And yes, as discussed in the second paragraph of section 3.1, the matrix for unary predicate is diagonal, i.e. the $(i,i)$th entry of $\mathbf{M}_k$ is 1, iff $P_k(\mathbf{x}_i)$ exists in the KB.
> > >
> > >
> > >
> > > - In equation (4), what is K? Where is this stated?
> > >
> > > $K$ denotes the total number of predicates, it's stated in the sentence above eq.1
> > >
> > >
> > >
> > >  - Shape of  $s_\psi^{(t)}$ and $s^{(t)}_{\varphi,k}$ in equation (4)
> > >
> > > Yes, $s_\psi^{(t)}$ and $s^{(t)}_{\varphi,k}$ are scalars. Thus the path attention vector $\mathbf{s}_\psi$ is of the shape $T$, and the operator attention matrix $\mathbf{S}_\varphi$ is of the shape $T\times K$. And $\kappa(\mathbf{s}_\psi, \mathbf{S}_\varphi)$ is a matrix of shape $|\mathcal{X}|\times |\mathcal{X}|$ where $|\mathcal{X}|$ is the total number of entities.
> > >
> > >
> > > - In equation (3), (4), what do the two Ts mean?
> > >
> > > We apologize for the confusion on the notations. We use superscript $T$ for variables such as vectors $\mathbf{v}_\mathbf{x}$ or stacked variables $[s_\psi^{(1)}, ..., s_\psi^{(T)}]$ only to denote transpose. When denoting the variable at $(T)$ step, we always use the superscript $(T)$, e.g. $s_\psi^{(T)}$. This is defined in the second paragraph of section 2.2, and it is repeated in the paragraph below eq(4) as well. Apart from these, $T$ used on top of $\prod$ and $\sum$ denotes the steps of $T$.

---

> ### Author Response · Authors · 2019-11-13
> **Response to Reviewer3 - Cont'd**
>
>
> -Is the logical operator parameterized and learned?
>
> Our model does not learn the logic operator itself, i.e. $\varphi$. It learns the weights that combine the operators for producing the results. We apologize for the confusion here. We have made significant modifications for this part (sec 3,4 and Figure 3,4) in our 2nd draft.
>
> Our model closely follows the setup in the multi-hop reasoning literature. It operates over a KB that contains a fixed set of predicates, where each of them is represented as a binary matrix. These matrices are fixed and never learned, because they encode the KB itself. The goal of the model is, when given a query $\langle x, P^*, x' \rangle$, to find a relational path that lead from $x$ to $x'$, where the path itself is parameterized as the chain of matrix multiplications on the one-hot vector of the input (as shown in eq3 of the 2nd draft). Intuitively, the model learns what matrix to multiply at each step that could lead to the correct results.
>
> In sec 3, we discuss how these hard-choices of picking matrices can be relaxed into taking the weighted sum over the matrices and over the scores (eq10,13 and Figure 3). So what our model learns is a set of attentions that guide the weighed sums. Therefore, even though the logical operators are parameterized into adjacency matrices, they are not learned by the model.
>
>
>
>
> -How do you derive the crisp rules shown in Table 4 in the appendix?
>
> Our model is split into two parts: rule generation (Figure 4) and rule evaluation (Figure 3). In the rule generation phase, we run our Transformer model and generate a set of attentions (described in sec 4). This encodes the soft picks of operators,  statements and the formulas. And this part alone does not generate any query-specific outputs. It only operates over a set of embeddings: embeddings of "dummy" input $\mathbf{e}_X$, $\mathbf{e}_{X'}$, and embeddings of predicates $\mathbf{H}$. All the computations in this phase are to "mimic" the actual evaluation that could happen when the true query is fed (which will happen in the next phase), and the goal is to generate the attentions along this simulated evaluation. After this phase,  the only things that go to the next phase (Figure 3) are the attentions $\mathbf{S}_\varphi, \mathbf{S}_\psi, \mathbf{S}_\psi', \mathbf{S}_f, \mathbf{S}_f'$ and $\mathbf{s}_o$.
>
> During training, we use attentions to perform weighted sums over the matrices to get the output score. For validation, testing and visualization (discussed in section 5), we simply take the argmax over these attentions, and that gives us the hard samples that the model picks. For example, for a single input $\mathbf{e}_X$ and two operators $[\varphi_1, \varphi_2]$. For one step of operator call search, we will obtain the attentions of shape 2X1, say it's $[0.8, 0.2]$. During training, we perform weighted sum $(0.8\mathbf{M}_1 + 0.2\mathbf{M}_2)\mathbf{v}_\mathbf{x}$ as shown in eq5. And for rule extraction, we take argmax and get $[1, 0]$, so we visualize the rule as "Phi_1(X)"
>
>
>
>
> -How do you ensure that gradients do not vanish?
>
> Our model is a stack of 3 Transformer networks each with a slight modification. Within each Transformer, there are residual connections for each attention operation and the following pointwise feed forward network. So the gradient does not vanish.

---

### Author Response · Authors · 2019-11-13
**Paper Revision**

We thank all reviewers for their comments and suggestions. We realized that the initial presentation was problematic in presenting the methodology part and was causing constant confusion during reading.  As suggested by the reviewers, we have made a major revision on the paper. We revised most of section 2,3,4 and 5, supplemented the introduction and related works section to reflect the literature, and added more baselines into the experiments.

In the 2nd draft:
	- We include an introduction for existing literature of multi-hop reasoning methods into section 2.
	- We draw a close connection between our model with the literature and have made the notations and presentations aligned.
	- In section 3, we introduce the operator, statement and formula space and how they are relaxed into weighted sums w.r.t attentions.
	- In section 4, we introduce the Transformer networks for generating these attention parameters.
	- In section 5, we introduce the training, and how to extract explicit rules
	- In experiments, we have added TransE and RotatE for KB completion task, and added a few low-accuracy example rules in the appendix.
	- In the related work, we have included a review on symbolic ILP, Neural Logic Machine, and link prediction methods.

Additionally, we have uploaded the main part of our implementation into the public repository https://github.com/gblackoutwas4/NLIL . Currently, we are still in the process of cleaning up the code, so there can be glitches in running the model. Hopefully, we will release the complete version in the next few weeks.

We apologize for the writing issues in the first draft. And we would greatly appreciate it if the reviewer can give another pass on the paper.  Many thanks!

---

### Public Comment · ~Juan_Recal1 · 2020-06-08
**independent review**

I came across this paper recently for it proposes an approach to inducing rules in a differentiable manner using attention maps generated using transformers. After spending time reading it several times, checking the code I thought I would provide an independent review and save others the hassle of getting lost. I would argue this paper should be accepted too but I’ll draw attention to these points for anyone reading the published version of this paper for their own sake of sanity:

1. From the very beginning you are lured into this paper with Figure 1 (the one where we see regions of images in a rule-like form). This figure is mainly eye-candy and misleading. Despite the example being motivated throughout with cars, and clothing, in Section 6.2 we realise that it actually runs on the pre-processed output of a pre-trained network that generates the knowledge base from the image. The labels, relations and objects are given, with final “KBs contain 213 predicates”. Hence the task on Figure 1 is misleading as it does not involve anything with images or learning what a car is insofar as the image region is concerned. The authors point out that the supervised method achieves better performance on the Visual Genome dataset since “it relies on highly informative visual features” suggesting their method is more disconnected from the actual image than Figure 1 implies. Does Figure 1 look fancy? Absolutely.

2. This paper suffers from what conferences refer to as “decorative math”. If you ask yourself what is going on mid-way through because you can’t remember which greek letter associated with some vectors and matrices do what is because most of it is decorative. Saying we compute attention and perform weighted sums with an informative example is too easy compared to laying out everything in fancy mathematical notation. Even better, the authors make the diagrams in Figure 3 and 4 include those equations and symbols in a picturesque form so in case you stopped following what was going on, you can look at the diagrams for more symbols. It gets so bloated that the authors “give unary $\psi$ a dummy input x for notational convenience.” The reviewers #3 (I find it difficult to read) and #2 (notation is utterly confusing) point this out and the authors respond by making “significant changes”. Significantly rewriting does not necessarily guarantee clarity. I would recommend moving all the mathematical details beyond a few key equations to the appendix. In case you decide to venture into the source code they provide, besides running the experiments, it has no comments or correspondence in how the paper is organised, so you can puzzle out which greek letter corresponds to which PyTorch module (More on source code later).

3. The usage of first order logic and its representation extends (2) into the terminology domain. The authors clearly come from a graph traversal background, in particular Gu et al 2015, but interleaving the main ideas they propose with FOL and ILP is coincidental and related as opposed to being critical. For example, the authors “introduce the notion of a primitive statement” which looks like just functions in FOL, p(f(X)). Indeed their description of “a typical” FOL does not include functions, just entities, predicates and formula which is obviously convenient when one considers graph traversal and KB completion tasks (i.e. nodes = entities, edges = predicates and paths = formula). Since they introduce functions with Skolem normal form, introducing a new terminology does not make sense from an FOL / ILP perspective.

4. To take point (3) further, it’s really weird that the authors ignored the Area Chair’s comment of “strongly, *STRONGLY*” advice to compare to traditional ILP systems and Neural Logic Machines which does have a stronger foundation on logic despite not producing rules. $\partial$ILP also kindles from FOL and ILP background. Yet retrofitting KB traversal into a ILP domain requires more careful thought which despite the warning from the reviewers seems to have slipped through. So from an ILP perspective I would take this paper with a lot of precaution. Section 2.1 on ILP has no references to define ILP and is left at the mercy of the authors interpretation of ILP conveniently fitting it for their approach. In particular, the area chair’s suggestion of comparing to traditional ILP and NLM is critical if one pre-processes the image with a pre-trained network to perform rule learning on images, in reality the system is just rule learning again on KBs. With the simplicity of the learnt rules in Visual Genome processed dataset (Table 4 in Appendix), symbolic learners such as ILASP, FastLAS that scales to very large hypothesis spaces, FOIL, Metagol and even Progol might indeed succeed.

---

> ### Public Comment · ~Juan_Recal1 · 2020-06-08
> **independent review part 2**
>
> 5. Going back to the exciting goal of learning to explain, the paper does not present the rules learnt for datasets other than Visual Genome pre-processed dataset. For a paper selling ideas of rule-learning as a form of explanation, we only get Table 4 and 5 (with some low accuracy rules) in the appendix (!) as the only rules learnt. We also don’t know the distribution of the relations, reviewer #3 points it out with authors responding we take argmax of [0.8 M1, 0.2M2] but this is a very different convergence point from [1.0 M1, 0.0 M2] relying on the fact that attentions “usually become highly concentrated”. If one was to read the main body of this paper, one would be baffled to not find any explanation in the empirical evaluation despite the title starting with “learn to explain”. You then notice that in Appendix C, for KB completion tasks the formula combinations are disabled (L=0) so that “complex formulas are not preferred”. To this lack of explanation (pun intended), let’s also add that we are not provided with error bars or how reproducible these rules are. To nail the ILP coffin, is this system sound or complete with respect to the rules it can learn? Noting that since the conversion in Eq. 8 is not unique, one begs the question how consistent are the rules learnt and thus the performance of the system. Figure 1, table 4 all seem cherry picked.
>
> 6. Let’s focus on the actual contribution and the only learnable component of this approach, the hierarchical transformer networks. As reviewer #2 points out, the authors clarify in the rebuttal that the relation operators (the Mk matrices) are not learned, but the attention maps which select the operators are. So in essence what we build up to after 2-3rds of the paper is a ground input free rule attention generator. This is why the paper is novel, generating rules without relying on the input query and using “dummy inputs” is a good idea. But who is going to decipher all that MultiHeadAttention and Transformer business. Put the decorative math aside, the authors do not make it any clearer that what they write as MultiHeadAttention is actually a full Transformer network (had to check the source code) with Layer norms and everything. Again a simple working example on how to select functions, relations and formula to then detail in the appendix would be a breather. What is wrapped up in a for loop and a Transformer module in the source code is turned into bloated, fancy formulations. How is the Mk constructed, what does an actual rule generation example look like, we are left in the dark still in the published version of this paper.
>
> 7. A crucial concept in ILP are negative examples which prune the search space to avoid trivial or too general rules. Reviewer #1 picks up on this and authors respond that negative sampling is now described in section 5. The authors mention that they “sample nonexistent queries” “uniformly from the target query matrix M* where the entry is 0.” This is brushed over for some reason, it means that if the dataset does not contain an entry then we assume its false, i.e. closed world assumption. These negative examples are explicit in ILP to constrain the rule search space and avoid trivial rules. The consequences of this step are vague and perhaps not as significant from KB completion perspective but they are in the domain of rule learning. Basically, creating a narrative on ILP, FOL and rule-learning over graph traversal tasks is not done with the attention it deserves. The authors rebuttal changes are just short sentences to “address” the reviewers comments but fall short realising the bigger picture.
>
> 8. Building on point number (5) on lacking learnt rules, it’s difficult to see how the evaluation / experiments in the paper analyse the properties of rules. Beating state-of-the-art is *not* the key focus here, and the approach not beating specific sota methods are as important to publish a novel approach. However, with that in mind we don’t get any analysis for the approach: the qualitative analysis of rules, whether they fit an explanation, are they always interpretable and is there an ablation study? How much of this machinery is necessary to learn all this? Since appendix C says for KB completion the complex rule is disabled and the fact that most of the rules presented in Table 4 and 5 do not include negation (the ones that do give low accuracy as state caption of Table 5), one is left to guess as to why the authors went about evaluating a system to learning rules and explanations in a purely quantitative fashion. Even creating a synthetic KB dataset to ensure their system is able to learn negated rules, rules of a certain length, rules with a certain number of existential variables and how consistently it performs when different parts of the network are ablated is a big void that us readers need to guess.

---

> > ### Public Comment · ~Juan_Recal1 · 2020-06-08
> > **independent review part 3**
> >
> > 9. Finally, publishing source code in a repository with no comments, no clear structure with a fast, sloppy implementation and no version history gives no assurance that it is extensible, reusable or even correct. The main function is called train2() which makes me wonder what happened to the first training function. Or the fact that there is an odd line left TODO with no description of what to do (https://github.com/gblackout/NLIL/blob/iclr2020/model/Models.py#L233) I would wish the best of luck for anyone who is asked to compare their work to this and run experiments against their model: either reimplement following fancy math in the paper (you will likely miss things like dropout etc) or figure out how to run their code or model on different data.
> >
> > Overall I hope this review highlights some of drawbacks of the paper despite its merits such that in the future people think more when reading this paper. To clarify, I have no affiliation with the reviewers, authors, the paper or the conference.

---

### Decision · Program_Chairs · 2019-12-19

**Decision:**

Accept (Poster)

**Comment:**

This paper proposes a differentiable inductive logic programming method in the vein of recent work on the topic, with efficiency-focussed improvements. Thanks the very detailed comments and discussion with the reviewers, my view is that the paper is acceptable to ICLR. I am mindful of the reasons for reluctance from reviewer #3 — while these are not enough to reject the paper, I would strongly, *STRONGLY* advise the authors to consider adding a short section providing comparison to traditional ILP methods and NLM in their camera ready.